# TreeGear: Learning Graph Edit Distance with Zero Ground-truth Labels

## Abstract

Graph Edit Distance (GED) is a fundamental measure for assessing similarity between graphs, with broad applications across domains such as bioinformatics, cheminformatics, and social network analysis. Unfortunately, computing exact GED is NP-hard. Besides a number of approximation algorithms, neural methods have emerged as a promising solution to this challenge. However, the training of these neural models requires a large number of ground-truth labels, which is computationally expensive to obtain due to the NP-hardness, thereby hindering their scalability. In this work, we introduce a novel framework, TreeGear for learning GED without the need of ground-truth GED labels. Our approach uses structural supervision from tree edit distances (TED), which can be computed in polynomial time, enabling the model to learn meaningful representations from approximate signals. Unlike existing approaches that directly regress to GED, TreeGear learns pairwise node mappability scores through node embeddings, on which, we apply a *neighbor-biased mapper* to derive the best possible edit paths between two graphs. This novel reformulation enables strong out-of-distribution generalization, interpretability, and better alignment with the properties of the true GED. Extensive experiments across GED benchmarks demonstrate that TreeGear achieves state-of-the-art results, beating both non-neural and neural baselines that are trained on 100% ground-truth GED. Moreover, TreeGear is architecture-agnostic and generalizes effectively to unseen graphs, making it suitable for real-world deployment across diverse graph domains.

## 1 Introduction and Related Works

Graph Edit Distance (GED) is a fundamental distance metric for graph data with important applications in cheminformatics (Garcia-Hernandez et al., 2019; Gaüzere et al., 2012), image analysis Liu et al. (2011); Zhang et al. (2016); Madi et al. (2017), and cybersecurity Bourquin et al. (2013); Zhang et al. (2014) among others. GED quantifies similarity between graphs and provide fine-grained structural comparison by modeling the minimal sequence of edit operations, such as node/edge insertion, deletion, and substitution, required to transform one graph into another. For example, in bioinformatics and cheminformatics, it is used for comparing molecular structures to identify functional similarities or differences (Ranjan et al., 2022).

Despite its applications, the practical adoption of GED is significantly hindered by its computational complexity. Computing exact GED is NP-hard, as it involves exploring all possible mappings between the nodes of two graphs to determine the minimal edit path Bai et al. (2019). It is also APX-hard Lin (1994), making even polynomial-time approximation algorithms infeasible.

**Existing frameworks and their limitations.** To address this computational bottleneck, a variety of heuristics have been proposed, spanning both non-neural and neural approaches. We point to Blumenthal et al. (2020) for a survey on all non-neural heuristics for GED. While these approaches are interpretable and provide the edit path associated with the approximated GED, the prediction accuracy and computational efficiency has been surpassed by the more recent generation of neural heuristics(Ranjan et al., 2022; Bai et al., 2019; Piao et al., 2023; Zhang et al., 2021; Wang et al., 2021; Zhuo & Tan, 2022; Jain et al., 2024; Bai et al., 2020; Doan et al., 2021; Li et al., 2019).

**Neural heuristics.** These models, typically built on top of GNNs, aim to learn meaningful graph representations that can be used to predict similarity scores or approximate edit paths. While promis-

ing, neural heuristics are inherently supervised and rely on a large volume of training data annotated with *exact* GED values, which are themselves NP-hard to compute. This reliance on NP-hard ground-truth supervision leads to some major limitations.

- **Expensive supervision and restriction to small graphs:** Since computing the exact GED is NP-hard, data generation is prohibitively expensive, spanning days or even weeks Ranjan et al. (2022). Owing to the same reason, existing neural models are typically trained only on small graphs, leading to poor scalability and degraded performance on larger instances.
- **Poor cross-domain generalization:** Current GED approximators struggle to generalize beyond the distribution of graphs seen during training, even within the same domain (such as molecule graphs). Their performance deteriorates sharply on graphs with larger sizes, higher node degrees, or more complex structures, forcing models to undergo dataset-specific training for each new domain. This repeated training pipeline is expensive due to the NP-hardness of solving GED.
- **Lack of interpretability:** Most neural approaches predict only the GED value without producing the corresponding edit path. The absence of edit paths limits interpretability in domains where understanding structural differences is essential, such as analyzing functional roles in protein complexes Singh et al. (2008), performing image alignment Conte et al. (2003), or uncovering gene regulatory mechanisms Chen et al. (2018). While a few methods (e.g., GEDGNN Piao et al. (2023), GENN-$A^*$ Wang et al. (2021)) attempt to provide interpretability, they often sacrifice scalability or generalizability.
- **Lack of fundamental guarantees:** Most neural GED approximators produce continuous predictions without guaranteeing that outputs respect core distance properties such as non-negativity, symmetry, identity of indiscernibles, or integral upper bounds. The absence of such guarantees undermines reliability in downstream applications like clustering, indexing, and similarity search.

**Contributions.** We mitigate the outlined gaps through the following core contributions.

- **Tree-Based Framework for GED Learning:** We introduce a novel label-efficient framework, TREEGEAR (Tree-based GED Estimation using Alignment and Representation), for learning GED without relying on ground-truth supervision. TREEGEAR leverages *ordered tree edit distance (*TED*)*—which can be computed in polynomial time—as a structural supervisory signal. By distilling knowledge from tree representations, our method captures rich structural semantics while avoiding the prohibitive cost of computing true graph edit distances.
- **Fully Label-Free GED Prediction:** In contrast to existing neural GED models that require extensive ground-truth annotations, our approach is entirely label-free. TREEGEAR dispenses with any dependence on GED labels, substantially reducing annotation costs and enabling deployment in real-world settings where such labels are scarce or infeasible to compute.
- **Out-of-Domain Generalization and Interpretability:** TREEGEAR generalizes seamlessly across graph domains without retraining or fine-tuning, scaling effectively to extraordinarily large graphs where true GED computation is intractable. This is achieved by a novel reframing of the neural approximation task: instead of directly regressing to the GED, the neural model predicts a *mappability score* for each pair of nodes across the two graphs, which quantifies how well the nodes align in terms of structure and attributes. These pairwise scores are then assembled into a weighted bipartite graph and bipartite matching on this graph yields the approximate GED and the corresponding edit path. This restructuring ensures invariance to graph size distributions, guarantees key distance properties (upper-bound validity, non-negativity, symmetry, and identity), and improves interpretability by explicitly revealing the edit operations implied by the alignment.
- **Excellence in Empirical Performance:** Our framework is architecture-agnostic and can be integrated into any GNN-based GED prediction pipeline. Extensive experiments demonstrate that TREEGEAR consistently outperforms both supervised neural and algorithmic baselines on standard benchmarks.

## 2 BACKGROUND AND PROBLEM STATEMENT

**Definition 1** (Graph). *An undirected graph with labeled nodes is denoted by $\mathcal{G}(\mathcal{V}, \mathcal{E}, \mathcal{L})$, where $\mathcal{V} = \{v_1, \ldots, v_{|\mathcal{V}|}\}$ represents the set of nodes, $\mathcal{E} \subseteq \mathcal{V} \times \mathcal{V}$ defines the edges, and $\mathcal{L} : \mathcal{V} \to \Sigma$ is a function that assigns a label from the set $\Sigma$ to each node.*

In the case of unlabeled graphs, each node is assigned the same default label.

**Definition 2** (Node Mapping). *Given two graphs $\mathcal{G}_1$ and $\mathcal{G}_2$ with $n$ nodes each, a node mapping is a bijection $\pi : \mathcal{V}_1 \to \mathcal{V}_2$, ensuring that every node $v \in \mathcal{V}_1$ corresponds uniquely to a node $\pi(v) \in \mathcal{V}_2$.*

**Handling Unequal Graph Sizes:** If the number of nodes differs across graphs, say $n_1 < n_2$, we extend the smaller graph $\mathcal{G}_1$ by appending $n_2 - n_1$ isolated nodes referred to as *dummy* nodes. These are assigned a distinct label $\epsilon$ to indicate their placeholder nature and remain disconnected from the rest of the graph. Henceforth, we assume all graphs under comparison have the same number of nodes, achieved by such padding if necessary.

**Definition 3** (Edit Distance for a Fixed Node Mapping). *Let $\mathcal{G}_1(\mathcal{V}_1, \mathcal{E}_1, \mathcal{L}_1)$ and $\mathcal{G}_2(\mathcal{V}_2, \mathcal{E}_2, \mathcal{L}_2)$ be two graphs, and let $\pi$ be a bijective mapping between their nodes. The edit distance under $\pi$ is computed as:*

$$\mathrm{GED}_\pi(\mathcal{G}_1, \mathcal{G}_2) = \sum_{v_1 \in \mathcal{V}_1} \mathbb{I}(\mathcal{L}_1(v_1) \neq \mathcal{L}_2(\pi(v_1)))$$
$$+ \frac{1}{2} \sum_{v_1 \in \mathcal{V}_1} \sum_{v_2 \in \mathcal{V}_1} \mathbb{I}(e_1(v_1, v_2) \neq e_2(\pi(v_1), \pi(v_2)))$$

*where:*
- $e_i(u, v) = 1$ *if edge* $(u, v)$ *exists in* $\mathcal{E}_i$*, otherwise* $0$.
- $\mathbb{I}(A)$ *is the indicator function that returns* $1$ *if the condition $A$ is true, and* $0$ *otherwise.*

**Understanding Edit Costs:** The first term in the expression penalizes label mismatches. These capture substitutions when two real nodes have different labels, and account for insertions or deletions when a dummy node (labeled $\epsilon$) is involved. The second term penalizes structural discrepancies: it identifies edges that are present in one graph but not in the mapped location of the other graph. The factor $\frac{1}{2}$ prevents double-counting of edge mismatches.

**Definition 4** (Graph Edit Distance (GED)). *The overall graph edit distance between $\mathcal{G}_1$ and $\mathcal{G}_2$ is the minimum edit cost taken over all possible node mappings:*

$$\mathrm{GED}(\mathcal{G}_1, \mathcal{G}_2) = \min_{\pi \in \mathcal{M}} \mathrm{GED}_\pi(\mathcal{G}_1, \mathcal{G}_2) \tag{1}$$

*Here, $\mathcal{M}$ denotes the set of all bijections between the node sets.*

An example of GED computation is shown in Fig. 1.

Computing the GED is computationally intractable for large graphs, as the number of possible node mappings grows factorially with the number of nodes ($|\mathcal{M}| = n!$ where $n = \max\{|\mathcal{V}_1|, |\mathcal{V}_2|\}$), making the problem NP-hard and APX-hard.

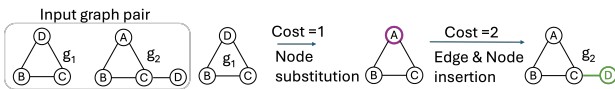

Figure 1: Example of GED computation. Here, GED = 3.

**Definition 5** (Tree Edit Distance (TED)). *Tree Edit Distance is a special case of Graph Edit Distance where both input graphs $\mathcal{G}_1$ and $\mathcal{G}_2$ are trees and the set of valid edit operations includes node deletion, node insertion, and node substitution (relabeling) where node deletion involves deleting a node as well as rewiring all of its children to their grandparent and node insertion is the complement of deletion. For ordered Tree Edit Distance, left-to-right orders among siblings are pre-defined and must be respected.*

**Ordered TED.** Ordered TED can be computed in polynomial time Zhang & Shasha (1989); Bille (2005). For conciseness, we will now refer to ordered TED as TED.

**Our Objective.** Existing neural approximators pose the GED learning problem as a regression task over a training set $\mathbf{Q}$, where each instance is of the form $\langle \mathcal{G}_1^i, \mathcal{G}_2^i, \mathrm{GED}(\mathcal{G}_1^i, \mathcal{G}_2^i) \rangle$, with the ground-truth GED as the supervision signal He & Singh (2006). This creates a fundamental paradox: in order to train a model to approximate an NP-hard problem, we first need supervision derived from solving that very same NP-hard problem—often via exhaustive or costly optimization procedures. This circular dependency raises a critical question: *can we design a neural model to approximate GED without relying on any supervision derived from NP-hard GED computations?*

To mitigate this limitation, we observe that GNNs operate by decomposing input graphs into sets of *computation trees*, where each node's embedding is recursively computed from its neighbors over a fixed number of message-passing steps. These local computation trees are then aggregated to form a holistic representation of the graph. This observation motivates a key question: *can a GNN be trained using supervision from TED instead of GED, and still effectively approximate GED?*

Training with TED supervision offers a significant advantage: it avoids reliance on ground-truth GED, thus sidestepping the need for NP-hard supervision. Furthermore, since GNNs already treat

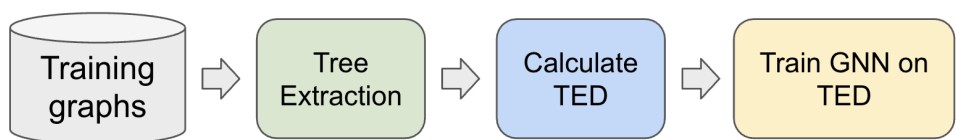

(a) Training with TED as proxy: For each input graph pair, we extract hybrid trees using a combination of label propagation and structurally faithful traversal. The TED between the trees is used as a proxy supervision to train a GNN. The GNN learns node-level representations that capture structural similarity through TED, bypassing the need for exact GED labels, which are NP-hard to compute.

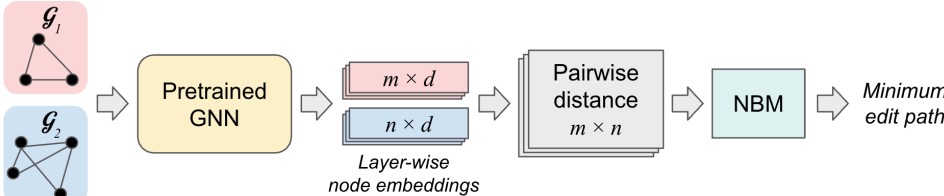

(b) Inference with Neighbor-biased Mapper (NBM): Given a trained GNN, we extract layer-wise node embeddings for a test graph pair. These embeddings are used to construct a distance matrix, which is then passed to the NBM to obtain a node alignment that respects both local similarity and structural consistency. The alignment is converted into a valid edit path whose cost approximates the GED. To tighten the upper bound, multiple alignments are generated using an ensemble of models and layers, and the one with the lowest edit cost is selected.

Figure 2: The main pipeline of TREEGEAR.

graphs as compositions of tree-structured computations, replacing GED with TED as the supervisory signal is unlikely to constrain the model's representational power. Empowered with these intuitions, we now formally state our problem as follows.

**Problem 1** (Learning to approximate GED with TED). *Let $Q$ be a training dataset where each instance is of the form $\langle \mathcal{T}_1^i, \mathcal{T}_2^i, \text{TED}(\mathcal{T}_1^i, \mathcal{T}_2^i) \rangle$ where $\mathcal{T}_1^i$ and $\mathcal{T}_2^i$ are trees. From $Q$, we aim to learn a GNN model $\Phi$ that enables us to approximate GED for any unseen pairs of graph $\mathcal{G}_1$ and $\mathcal{G}_2$ well. Mathematically, learn a function $f : (\mathcal{G}_1, \mathcal{G}_2, \Phi) \to \mathbb{Z}^+$ that takes as input a graph pair $\langle \mathcal{G}_1, \mathcal{G}_2 \rangle$ and model $\Phi$, and outputs a non-negative integral distance that minimizes:*

$$|f(\mathcal{G}_1, \mathcal{G}_2, \Phi) - \text{GED}(\mathcal{G}_1, \mathcal{G}_2)| \tag{2}$$

Note that in our formulation, the GNN $\Phi$ does not directly predict the GED. Rather, we aim to pass the output of $\Phi$ through another function $f$ that maps the GNN's output to our GED prediction.

## 3 PROPOSED METHODOLOGY: TREEGEAR

TREEGEAR addresses the limitations of existing GED prediction models. To eliminate the need for costly ground-truth GED labels, TREEGEAR uses a weakly supervised training strategy that leverages tree edit distance (TED) as proxy supervision. Specifically, for each pair of training graphs, we extract a pair of trees and compute their edit distances which can be computed in polynomial time. At inference time, we use our trained model to generate node embeddings, and subsequently perform node alignment using the Neighbor-biased Mapper (NBM). Among all the valid edit paths—one for every node alignment plan—we obtain the one with the smallest edit costs. The pipeline of TREEGEAR is shown in Figure 2.

### 3.1 TRAINING WITH TREE PAIRS: FROM GRAPHS TO TREES

As outlined in our objective our primary objective is to reduce the labeling cost without compromising the predictive performance of the base GNN-based model for GED prediction. To this end, we adopt ordered Tree Edit Distance (TED) as a proxy for Graph Edit Distance (GED). This choice is motivated by two key advantages: (i) ordered TED can be computed efficiently in polynomial time Zhang & Shasha (1989), and (ii) the message-passing architecture of GNNs inherently decomposes graphs into computation trees Gupta et al..

For TED to provide a high-quality approximation of GED, it is essential that the selected pair of trees cover the graph regions where structural differences (i.e., edits) are most likely to occur. Achieving this requires two conditions: (1) the neighborhoods of the roots from which the trees are expanded must intersect with these regions of interest, and (2) the extracted trees must preserve the structural

Figure 3: The tree extraction module in TREEGEAR.

semantics relevant to a GNN's computation. To meet these requirements, we propose a principled tree generation strategy that ensures alignment with both the underlying graph structure and the GNN's computational view. The pipeline of our tree generation module is illustrated in Figure 3.

### 3.1.1 ROOT SELECTION AND ALIGNMENT

To identify root nodes for the hybrid trees such that they best capture potential edit regions, we employ a label propagation strategy inspired by the Weisfeiler-Lehman (WL) test Weisfeiler & Leman (1968). We use these embeddings to perform soft node alignment between two graphs.

**Label propagation update rule.** We define the update rule for label propagation at iteration $t$ as:

$$z_t^i(u) = z_{t-1}^i(u) + \frac{1}{\mathbf{d}} \sum_{v \in \mathcal{N}(u)} z_{t-1}^i(v), \quad \forall i \in [1, l] \tag{3}$$

where: $z_t^i(u) \in \mathbb{R}$ denotes the $i$-th dimension of node $u$'s embedding at iteration $t$, $\mathcal{N}(u)$ is the 1-hop neighborhood of $u$, $l$ is the dimensionality of the one-hot encoding of node label $\mathcal{L}(u)$, and $\mathbf{d} \in \mathbb{R}_+$ is a discounting factor. This mechanism aggregates label information from the local neighborhood in a similar way as in the WL hashing, but takes into account the nodes' appearance order.

**Lemma 1.** *The label propagation update defined in Equation 3 ensures that:*

1. *For all $\mathbf{d}$, if $z_T(u) \neq z_T(v)$ then the rooted $T$-hop neighborhoods of the nodes $u$ and $v$ are non-isomorphic.*
2. *For $\mathbf{d} > M^T$ where $M = \max_{u \in \mathcal{V}} \deg(u)$, if $z_t(u) \neq z_t(v)$ then $z_{t+1}(u) \neq z_{t+1}(v) \ \forall t < T$.*

*Proof Sketch.* Since the contribution from neighbors is scaled by $\frac{1}{\mathbf{d}}$, any differences in the multiset of labels in $\mathcal{N}(u)$ and $\mathcal{N}(v)$ persist over iterations. The growth rate of $\mathbf{d}$ dominates, preserving uniqueness. A detailed proof using induction is provided in Appendix A.1.1. Note that the magnitude of $\mathbf{d}$ does not necessarily have a perfect correlation with the probability of hash collision.

**Root pair selection.** We select root pairs $(u^\star, v^\star)$ for a graph pair $(\mathcal{G}_1, \mathcal{G}_2)$ via the following procedure in four steps: (i) *Embedding computation:* We first run the label propagation for $T$ iterations to compute node embeddings $\{z_T(u)\}_{u \in \mathcal{V}_1}$ and $\{z_T(v)\}_{v \in \mathcal{V}_2}$. (ii) *Distance matrix:* We then construct matrix $D \in \mathbb{R}^{|\mathcal{V}_1| \times |\mathcal{V}_2|}$ such that: $D(u, v) = \|z_T(u) - z_T(v)\|$. Next, we perform the optimal assignment and obtain the final root. (iii) *Optimal assignment:* We use the Hungarian algorithm to solve: $\min_{\pi: \mathcal{V}_1 \to \mathcal{V}_2} \sum_{u \in \mathcal{V}_1} D(u, \pi(u))$. (iv) *Root extraction:* Lastly, we extract the root as follows:

$$(u^\star, v^\star) = \arg \min_{(u, \pi(u))} D(u, \pi(u)) \quad \text{subject to } D(u, \pi(u)) > 0$$

If all $D(u, \pi(u)) = 0$, we pick any $(u, \pi(u))$ arbitrarily. The main intuition is that the node pair $(u^\star, v^\star)$ selected via minimal embedding distance is likely to reside in regions with minimal structural divergence. Consequently, the rooted neighborhoods around $u^\star$ and $v^\star$ are most informative for approximating the graph edit path.

### 3.1.2 HYBRID TREE CONSTRUCTION

Here, we describe our hybrid tree construction procedure. Let $r \in \mathcal{V}$ be the root node selected using the alignment scheme from Section 3.1.1. Our goal is to construct a rooted tree $\mathcal{T}_r = (\mathcal{V}_\mathcal{T}, E_\mathcal{T})$, which we refer to as a *Hybrid Tree*. This tree is designed to preserve critical structural information from the original graph while remaining suitable for efficient computation of TED.

**Hybrid Tree.** A Hybrid Tree is an acyclic structure derived from $\mathcal{G}$ that satisfies the following: (i) Each original edge in $\mathcal{E}$ is included exactly once in $E_\mathcal{T}$; (ii) Duplication of a node occurs only when revisiting that node would form a chordless cycle; (iii) For any node that is not duplicated, its $k$-hop neighborhood in $\mathcal{T}_r$ is identical (as a multiset) to that in $\mathcal{G}$.

Algorithm 1 (Appendix A.1.2) outlines how to construct a hybrid tree from a graph, rooted at a given node. The goal is to approximate the graph's structure in a way that preserves its semantics for a GNN. Since each edge is traversed exactly once, the immediate neighborhood of every original (non-duplicating) node is retained, while dummy duplicating nodes are introduced only when closing cycles. As shown in Fig. 4, this ensures that the $k$-hop neighborhood multisets—used by message-passing GNNs —remain intact for all original nodes. Thus, from a GNN 's perspective, the hybrid tree differs from the original graph only in terms of duplicating nodes.

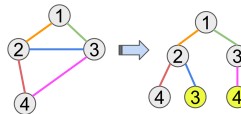

Figure 4: A hybrid tree of a graph. Duplicate nodes are highlighted.

**Lemma 2.** *In a hybrid tree, the number of duplicating nodes is equal to the number of chordless cycles in the original graph.*

The formal proof is given in A.1.2. On the other hand, for a BFS tree, the $k$-hop neighborhoods of all nodes in cycles of length $\leq k + 1$ are altered. For a computation tree, none of the nodes in the graph will preserve their $k$-hop neighborhoods for any $k \geq 2$. Furthermore, if we need to visit all nodes in the graph at least once and assume nothing about its connectivity, the expected size of the computation tree is at least: $\sum_{i=0}^{R} \bar{d}^i = O(\bar{d}^R)$; where $R$ is the graph's radius (the minimum eccentricity) and $\bar{d}$ is the average degree. This large size might make it computationally expensive and we can no longer take advantage of the efficiency of TED computation. We provide more details about the different tree types in Appendix A.1.2 and empirically evaluate their impact on performance in the ablation study (Sec. 4.2).

**Comparison of tree structures.** Let $\mathcal{G}$ have the average degree $\bar{d}$ and radius $R$:

- A *computation tree* of depth $R$ has the size of $\mathcal{O}(\bar{d}^R)$ due to the recursive addition of neighbors.
- A *BFS tree* has size $\mathcal{O}(|\mathcal{V}|)$, but distorts neighborhoods in graphs with cycles.
- A *Hybrid Tree* has size $\mathcal{O}(|\mathcal{V}|+c)$, where $c$ is the number of chordless cycles in $\mathcal{G}$, and it preserves neighborhoods for all original (non-duplicated) nodes.

**Dataset construction with TED.** Given a training graph pair $(\mathcal{G}_i^1, \mathcal{G}_i^2)$, we extract their Hybrid Trees $\mathcal{T}_i^1$ and $\mathcal{T}_i^2$, and compute their TED using the Zhang–Shasha algorithm Zhang & Shasha (1989) with arbitrary orders of siblings. As TED is only used to distantly supervise the GNN, the choice of sibling orders should have limited impact on the final matching quality, which depends on the relative ranking of node pairs rather than their precise similarity scores (see 4.2). The proxy dataset containing tree pairs and corresponding TEDs is: $\mathcal{D}_{\mathcal{T}} = \left\{ \left( \mathcal{T}_i^1, \mathcal{T}_i^2, \text{TED}(\mathcal{T}_i^1, \mathcal{T}_i^2) \right) \right\}_{i=1}^{n}$. To improve robustness and increase the diversity of node embeddings for use during inference, we train an ensemble of five GNN models. Each model is trained on a different half-sized subsample of $\mathcal{D}_{\mathcal{T}}$. These embeddings are later used for node alignment and GED estimation in Section 3.2.

### 3.2 INFERENCE WITH NEIGHBOR-BIASED MAPPER

Let $\mathcal{G}_1 = (\mathcal{V}_1, \mathcal{E}_1)$ and $\mathcal{G}_2 = (\mathcal{V}_2, \mathcal{E}_2)$ be two node-labeled graphs with associated embeddings $\{z(u)\}_{u \in \mathcal{V}_1}$ and $\{z(v)\}_{v \in \mathcal{V}_2}$ computed by a GNN trained with the TED-based proxy supervision (see Section 3.1.2). Our goal is to construct a node alignment $\pi : \mathcal{V}_1 \rightarrow \mathcal{V}_2$ that facilitates computation of a GED upper-bound. We describe the neighbor-biased mapper algorithm designed for this task.

#### 3.2.1 THE NEIGHBOR-BIASED MAPPER

The *Neighbor-Biased Mapper (NBM)* He & Singh (2006) is an iterative alignment algorithm that performs soft matching based on local embedding similarity and structural coherence. It operates on the pairwise distance matrix $D \in \mathbb{R}^{|\mathcal{V}_1| \times |\mathcal{V}_2|}$, where $D(u, v) := \|z(u) - z(v)\|_2 \ \forall u, v \in \mathcal{V}_1 \times \mathcal{V}_2$.

With the distance matrix, NBM creates a node alignment plan as follows. *(i) Pair selection:* First, we identify the closest pair of nodes $u, v$ such that $D(u^\star, v^\star) = \min_{x \in \mathcal{V}_1, y \in \mathcal{V}_2} D(x, y)$. *(ii) Neighbor biasing:* For every pair of neighbors $(w_1, w_2) \in \mathcal{N}(u^\star) \times \mathcal{N}(v^\star)$, we update the distance matrix by decreasing the corresponding entries: $D(w_1, w_2) \leftarrow D(w_1, w_2) - \delta$; where $\delta > 0$ is a hyperparameter controlling the bias strength. *(iii) Update:* Next, we add $(u, v)$ to the final result and remove the row of $u$ and the column of $v$ from $D$. *(iv) Repeat:* We repeat the process until $D$ is empty.

**Interpretation and robustness.** The key step here is the neighbor biasing (Step 2), which encourages local structural consistency by promoting the alignment of nodes whose neighbors have already been aligned. This approximates a greedy solution to a structure-aware node matching problem. Un-

like hard alignment approaches, NBM only depends on the relative ordering of pairwise distances and not their absolute values. Thus, with a suitable $\delta$, if the TED-trained embeddings preserve the ordering induced by the true GED-optimal mapping, the algorithm yields competitive approximations even under a scale shift of the node embeddings. We show its empirical evidence in Sec. 4.2.

### 3.2.2 GED COMPUTATION: EDIT PATH FROM NODE MATCHING

We now describe a procedure to derive an upper-bound edit path from $\mathcal{G}_1$ to $\mathcal{G}_2$ using $\pi$. Our strategy follows Piao et al. (2023) and leverages the fact that $\pi$ is derived from node embeddings trained using TED, as described in Section 3.1. We exploit these embeddings across multiple layers and models to generate several candidate alignments.

**Step 1: Node substitution.** For each $u \in \mathcal{V}_1$, if $\mathcal{L}_1(u) \neq \mathcal{L}_2(\pi(u))$, we substitute the label of $u$ with $\mathcal{L}_2(f(u))$. Let $c_s$ denote the cost of node substitution.

**Step 2: Edge deletion.** For each edge $(u, v) \in \mathcal{E}_1$, if $(\pi(u), \pi(v)) \notin \mathcal{E}_2$, we delete edge $(u, v)$. Let $c_d$ denote the cost of edge deletion.

**Step 3: Node insertion.** Let $W = \mathcal{V}_2 \setminus f(\mathcal{V}_1)$ (assuming that $|\mathcal{V}_1| \leq |\mathcal{V}_2|$). For each $w \in W$, we introduce a dummy node $w' \in \mathcal{V}_1'$ and set $\mathcal{L}_1(w') := \mathcal{L}_2(w)$. $c_n$ denotes the cost of node insertion.

**Step 4: Edge insertion.** Let $\pi_b : \mathcal{V}_1' \to \mathcal{V}_2$ be the extended bijective mapping and $\pi_b^{-1}$ its inverse. For each edge $(x, y) \in \mathcal{E}_2$, if $(\pi_b^{-1}(x), \pi_b^{-1}(y)) \notin \mathcal{E}_1'$, we insert the edge $(\pi_b^{-1}(x), \pi_b^{-1}(y))$ into $\mathcal{E}_1'$. Let $c_e$ denote the cost of edge insertion.

**Total edit path cost.** The total cost of the edit path $\mathcal{P}_\pi$ induced by the alignment $\pi$ is given by (with $\mathbb{I}[\cdot]$ being the indicator function):

$$\text{Cost}(\mathcal{P}_\pi) = c_s \sum_{u \in \mathcal{V}_1} \mathbb{I}[\mathcal{L}_1(u) \neq \mathcal{L}_2(\pi(u))] \ + \sum_{(u,v) \in \mathcal{E}_1 \setminus \pi^{-1}(\mathcal{E}_2)} c_d + \sum_{w \in W} c_n \ + \sum_{(x,y) \in \mathcal{E}_2 \setminus \pi_b(\mathcal{E}_1')} c_e$$

**Observation 1.** *The constructed path $\mathcal{P}_f$ transforms $\mathcal{G}_1$ into a graph $\mathcal{G}_1'$ that is isomorphic to $\mathcal{G}_2$. Therefore, $GED(\mathcal{G}_1, \mathcal{G}_2) \leq Cost(\mathcal{P}_\pi)$ by definition.*

**Final path selection.** Each necessary edit—substitution, insertion, deletion—is explicitly accounted for based on $\pi$, and after applying all edits, $\pi_b$ becomes a graph isomorphism. To reduce variance and improve approximation quality, we generate a set of candidate alignments $\{\pi^{(j)}\}_{j=1}^m$ using: (i) multiple TED-trained GNN models (3.1.2) and (ii) layer-wise embeddings from each model. For each alignment $\pi^{(j)}$, we compute the associated path cost $\text{Cost}(\mathcal{P}_{\pi^{(j)}})$ and select the optimal one: $\hat{\pi} = \arg\min_j \text{Cost}(\mathcal{P}_{\pi^{(j)}})$. The resulting path cost provides our final approximation to $GED(\mathcal{G}_1, \mathcal{G}_2)$, completing our weakly supervised GED inference pipeline.

## 4 EXPERIMENTAL RESULTS

In this section, with extensive experiments, we demonstrate that TREEGEAR achieves competitive or superior performance compared to models trained with full GED supervision. Our code is available at: https://anonymous.4open.science/r/ged_distill-B177/.gitignore.

**Datasets.** We use five benchmark datasets to comprehensively evaluate TREEGEAR. A detailed description of these is included in App. A.2.1. Table 5 in the Appendix provides a summary.

**Baselines.** We evaluate TREEGEAR against several recent state-of-the-art baselines, including GREED Ranjan et al. (2022), GEDGNN Piao et al. (2023), ERIC Zhuo & Tan (2022), GRAPHEDX Jain et al. (2024), and H2MN Zhang et al. (2021). We exclude older methods such as SIMGNN, GRAPHOTSIM, GMN, GRAPHSIM, TAGSIM, and GENN-A*, as they have been consistently outperformed by more recent models like GREED, GRAPHEDX, GEDGNN, and ERIC in prior benchmarks. Among non-neural baselines, we incorporate the top-performing heuristics identified in the benchmarking study by Blumenthal et al. (2020)–specifically: LP-GED-F2, COMPACT-MIP, ADJ-IP, BRANCH-TIGHT, NODE, and IPFP–as well as the more recent GEDGW by Cheng et al. (2025). For the backbone GNN model to train with TED, we use GREED, though our framework is compatible with any architecture that produces node embeddings. We evaluate generalizability of TREEGEAR to other GNNs in Appendix A.3.2.

**Metrics.** We use the Root Mean Square Error (RMSE) to assess the performance of the methods. In addition, we include the accuracy score— known as the Exact Match Ratio (EMR)—which measures the percentage of the perfect matches between the predictions and the true GED values.

## 4.1 RESULTS FOR GED PREDICTION

Our main results on RMSE are shown in Table 1. Our findings highlight three major takeaways: **First,** TREEGEAR consistently achieves the lowest RMSE on four out of five datasets. This demonstrates the effectiveness of TED as a proxy supervision signal and the robustness of our inference pipeline. Particularly, on IMDB, TREEGEAR reduces the RMSE by more than 75% compared to GREED (1.24 vs. 5.24), a fully supervised model and on code2, TREEGEAR outperforms all baselines, including neural and non-neural models, by a significant margin. In addition to RMSE, we evaluate the EMR in Table 2, which measures the percentage of predictions that exactly match the ground-truth GED. TREEGEAR sets a new state-of-the-art EMR on AIDS (88.8%), LINUX (99.5%), and IMDB (97.1%). Even on

Table 1: The RMSE (lower is better) of the methods on five benchmark datasets. The results from three different training seeds are reported in the format: $\mu \pm \sigma$ (the maximum standard deviation in TREEGEAR is .17 and baselines is .2). The best result for each dataset is highlighted. Our proposed TREEGEAR achieves best results in most cases.

| | | AIDS | LINUX | IMDB | code2 | molhiv |
|---|---|---|---|---|---|---|
| | TREEGEAR | 0.43 | 0.15 | 1.24 | 3.67 | 2.5 |
| Neural | GREED | 0.72 | 0.49 | 5.24 | 5.27 | 2.2 |
| | GEDGNN | 0.92 | 0.29 | 4.43 | 16.68 | 1.75 |
| | ERIC | 1.08 | 0.30 | 42.44 | 17.55 | 3.56 |
| | H$^2$MN | 1.14 | 0.6 | 57.8 | 11.96 | 12.01 |
| | GRAPHEDX | 0.78 | 0.27 | 32.36 | 21.46 | 14.14 |
| Non-Neural | ADJ-IP | 0.85 | 0.5 | 42.18 | 14.94 | 10.21 |
| | Node | 2.71 | 1.24 | 61.03 | 8.34 | 4.97 |
| | LP-GED-F2 | 1.96 | 0.23 | 55.26 | 16.03 | 12.86 |
| | Branch | 3.31 | 2.45 | 7.36 | 12.64 | 9.86 |
| | Compact-MIP | 2.69 | 0.44 | 65.88 | 19.46 | 10.88 |
| | IPFP | 4.18 | 2.29 | 69.45 | 15.19 | 13.69 |
| | GEDGW | 1.88 | 1.87 | 2.31 | 5.81 | 6.31 |

molhiv, where our RMSE is slightly higher than GREED, TREEGEAR achieves a better EMR (24.7% vs. 21.1%), indicating stronger top-rank accuracy.

**Second,** TREEGEAR generalizes well to unseen graphs, maintaining strong performance when trained on one dataset and tested on another (Fig 5a), outperforming GREED by a huge margin. To further stress-test its scalability, we evaluate TREEGEAR on the large ogb-ppa graphs (avg. 2K+ edges), where ground-truth GED is infeasible, making supervised methods unusable. However, since TREEGEAR guarantees an upperbound, we compare its values

Table 2: The EMR (higher is better) of the neural methods: Results from three different training seeds are reported (the maximum standard deviation in TREEGEAR is .7% and baselines is .9%).

| | AIDS | LINUX | IMDB | code2 | molhiv |
|---|---|---|---|---|---|
| TREEGEAR | 89% | ≈100% | 97% | 16% | 25% |
| GREED | 53% | 71% | 16% | 8% | 21% |
| ERIC | 58% | 79% | 17% | 9% | 23% |
| GEDGNN | 35% | 85% | 7% | 1% | 57% |

(trained on code2) with non-neural methods. Fig 5b shows that TREEGEAR achieves significantly tighter upper bounds than algorithmic baselines. **Third,** TREEGEAR achieves these results without access to ground-truth GED labels, requiring only TED supervision, which is orders of magnitude faster to compute (Table 6, Appendix A.3.1). These results demonstrate that TREEGEAR achieves high accuracy, label efficiency, and broad generalizability. It not only surpasses supervised neural models with 100% ground-truth labels but also outperforms established algorithmic baselines.

## 4.2 ABLATION STUDY

**Impact of different tree types.** To evaluate the effect of different tree construction strategies on both performance and computational efficiency, we compare the hybrid tree design in TREEGEAR with BFS trees and computation trees of GNNs. As described in Section 3.1.2, hybrid trees are designed to preserve the k-hop neighborhood structure of nodes. Table 4 reports the RMSE and average TED computation time for each tree type across three datasets. We observe the followings: (i) Hybrid Trees consistently outperform BFS trees, particularly on larger graphs such as those in the code2 dataset. This is attributed to the hybrid tree's retention of all edges, which ensures structurally faithful GNN message

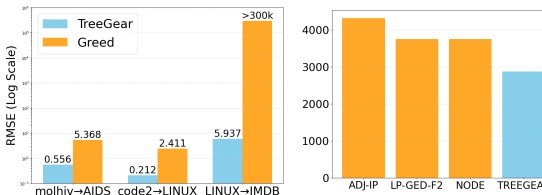

(a) Gener. across datasets  (b) Upperbound on ogb-ppa

Figure 5: Out-of-distribution generalization. In Fig 5a, for $A{\rightarrow}B$, we train on dataset $A$ but evaluate on $B$. TREEGEAR has much better performance than GREED. In Fig 5b, TREEGEAR trained on code2 yields the lowest GED upperbounds for ogb-ppa graphs.

Table 4: Comparison of different tree traversals in terms of RMSE and TED computation time. The results for Computation Tree is not available for code2 because its take huge time to compute the TED. For the available results, the computation time of TED is measured in ms/tree-pair.

| | AIDS | | LINUX | | code2 | |
|---|---|---|---|---|---|---|
| | RMSE | TED time | RMSE | TED time | RMSE | TED time |
| BFS Tree | 0.74 | 3.53 | 0.285 | 2.86 | 3.84 | 24.78 |
| Computation Tree | 0.48 | 8.52 | 0.058 | 9.23 | n/a | n/a |
| Hybrid Tree (TREEGEAR) | 0.43 | 3.51 | 0.151 | 2.91 | 3.67 | 34.00 |

propagation. (ii) While BFS trees are computationally cheaper due to their smaller size, this comes at the cost of reduced accuracy. (iii) On the other hand, Computation Trees become impractical for large graphs due to exponential growth in size (e.g., code2). They only slightly outperform hybrid trees (of TREEGEAR) on the LINUX dataset but at a significantly higher computational cost. Overall, the hybrid tree in TREEGEAR strikes the best balance between scalability and performance.

**Robustness of the NBM component.** We evaluate the robustness of TREEGEAR's NBM module (Sec. 3.2.1) to noise and distribution shifts in the learned embeddings by comparing it against the Hungarian matcher under two training regimes: node embeddings generated using (i) TED supervision (default of TREEGEAR), (ii) ground-truth GED. In Table 3, the performance gain from switching to GED-trained embeddings is substantially smaller for NBM than for Hungarian, indicating NBM's resilience to nominal embedding discrepancy and better generalization from weak supervision. Across all datasets, NBM achieves lower RMSEs and smaller performance shifts. Notably, on IMDB, the RMSE drop for NBM is only 0.66, compared to 3.46 for Hungarian. This stability stems from NBM's local matching strategy, which is less sensitive to global embedding distortions as long as the correct ranking of node pairs

Table 3: Improvement (in RMSE) of NBM and Hungarian when switching from TED-based to ground-truth GED-based embeddings. TREEGEAR uses TED and NBM. NBM, producing a smaller improvement, is more robust to noise and distribution shifts.

| | AIDS | IMDB | code2 |
|---|---|---|---|
| TED + NBM | 0.43 | 1.24 | 3.67 |
| GED + NBM | 0.37 | 0.58 | 3.42 |
| Improvement | 0.06 | 0.66 | 0.25 |
| TED + Hungarian | 3.61 | 7.33 | 20.83 |
| GED + Hungarian | 3.26 | 3.87 | 18.89 |
| Improvement | 0.35 | 3.46 | 1.94 |

is preserved. These findings support our hypothesis (Sec. 3.2.1) that NBM's inductive bias enables high-quality alignments under imperfect supervision. Appendix A.3.3 analyzes how the embedding budget (number of unique sets) affects the tightness of NBM-derived upperbounds.

## 4.3 INPUT SIZE VS RMSE

To further understand the robustness of TREEGEAR, we analyze how its prediction accuracy varies with the size of the input graphs. This is particularly relevant given that neural GED models often exhibit degraded performance on larger graphs. Figures 6a and 6b show the RMSE of TREEGEAR across different graph sizes for the IMDB and Code2 datasets, respectively. In both cases, TREEGEAR maintains a consistently lower RMSE relative

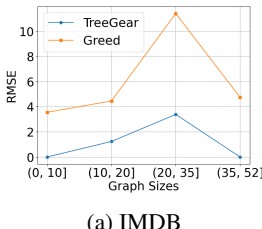

(a) IMDB

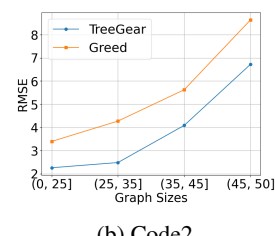

(b) Code2

Figure 6: RMSE variation by graph size (measured by number of nodes) on IMDB and code2 datasets.

to the fully supervised baseline, demonstrating its ability to scale effectively with graph size. We attribute TREEGEAR's overall stability to the design of the NBM, which operates predominantly at the node level. This node-centric design makes the inference process less sensitive to the global size of the graph, enhancing generalization to larger instances.

## 5 CONCLUSIONS

We introduced TREEGEAR, a scalable and supervision-efficient framework for learning graph edit distance (GED) by using tree edit distance (TED) as a proxy. By leveraging polynomial-time TED computations within a principled training and inference pipeline, TREEGEAR eliminates the need for costly ground-truth GED labels and addresses a long-standing challenge in neural GED approximation. Our approach achieves state-of-the-art performance across multiple benchmarks, surpassing both supervised neural models and classical heuristics, while producing interpretable edit paths and cross-domain generalization.

## REPRODUCIBILITY STATEMENT

To support reproducibility, we release our code at: https://anonymous.4open.science/r/ged_distill-B177/.gitignore. Besides the setup described in Section 4, additional details on datasets and hyper-paramter configuration are provided in Appendix A.2. All proofs of theoretical claims are given in Appendix A.1. In addition, we provide the pseudocodes in Appendix A.1.2.

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

# A  APPENDIX

## A.1  ADDITIONAL DETAILS OF OUR FRAMEWORK TREEGEAR

### A.1.1  LABEL PROPAGATION FOR ROOT SELECTION

**Re-stating the details of Lemma 1**

(1) **[Lemma 1.1]** The label propagation update defined in Equation 3 ensures the following: if the rooted $T$-hop neighborhoods of the nodes $u$ and $v$ are isomorphic, $z_T(u) = z_T(v)$. Its contrapositive also holds: if $z_T(u) \neq z_T(v)$, the rooted $T$-hop neighborhoods of the nodes $u$ and $v$ are non-isomorphic. This is true for any $\mathbf{d}$ used in Equation 3.

(2) **[Lemma 1.2]** With $\mathbf{d} > M^T$ where $M = \max_{u \in \mathcal{V}} \deg(u)$, it's guaranteed that: if $z_t(u) \neq z_t(v)$ then $z_{t+1}(u) \neq z_{t+1}(v) \ \ \forall t < T$.

**Proof of Lemma 1.1**

**Definition 6** (Rooted $T$-hop neighborhood). *The $T$-hop neighborhood rooted at node $u$ is the tuple: $(H_T(u), u)$ where $H_T(u)$ is the induced subgraph containing $u$ and the set of neighbors within $T$ hops from $u$. $\mathcal{V}(H_T(u)) = \{u\} \cup \bigcup_{1 \leq t \leq T} \mathcal{N}_t(u)$.*

**Definition 7** (Rooted $T$-hop neighborhood isomorphism). *$(H_T(u), u) \cong (H_T(v), v)$ if there exists a bijective mapping $\pi$ such that $\pi(u) = v$ and under $\pi$: $H_T(u) \cong H_T(v)$.*

It should be noted that the end goal of the label propagation procedure is to select roots for tree expansion and TED computation. Thus, Definition 7 is important because given two isomorphic graphs, if the pair of roots are misaligned, the resulting TED can be greater than $0$, potentially misleading the model.

We would like to remind the reader that $\mathcal{L} : \mathcal{V} \to \Sigma$ is a function that assigns a label from the set of labels $\Sigma$ to each node. As an extension, $\mathcal{L}(S) = \{\mathcal{L}(u) | u \in S\}$ where $S$ is a multiset of nodes in $\mathcal{V}$.

**Corollary 1.1**  If $(H_T(u), u) \cong (H_T(v), v)$, the multisets of the labels of $u$'s and $v$'s $t$-hop neighbors must be equal, i.e. $\mathcal{L}(\mathcal{N}_t(u)) = \mathcal{L}(\mathcal{N}_t(v)) \ \forall t \leq T$, and the labels' degrees must be equal. A label's degree is the degree of the node having that label.

*Proof.* For every node $w \in \mathcal{N}_t(u)$ with degree $\deg(w)$, there is a simple path $P$ of length $t$ from $u$ to $w$ in $H_T(u)$. Because $\pi(u) = v$ in the isomorphic mapping between $H_T(u)$ and $H_T(v)$, there must be a *bijective* mapping $\pi_{\mathfrak{p}}$ between the two sets of simple paths in $H_T(u)$ and $H_T(v)$ such that: $\mathcal{L}(P_j) = \mathcal{L}(\pi_{\mathfrak{p}}(P)_j)$ and $\deg(P_j) = \deg(\pi_{\mathfrak{p}}(P)_j) \forall j \in [1 : t]$. Therefore, for every label $\mathcal{L}(w) \in \mathcal{L}(\mathcal{N}_t(u))$, there is a bijectively mapped label of the same degree and value in $\mathcal{N}_t(v)$. $\quad\square$

**Corollary 1.2**  If $\mathcal{L}(\mathcal{N}_t(u)) = \mathcal{L}(\mathcal{N}_t(v)) \ \forall t \leq T$, then $z_T(u) = z_T(v)$ given the update rule defined in Equation 3.

*Proof.* $z_T(u)$ and $z_T(v)$ are vectors. Corollary 1.1 trivially extends to $\mathcal{L}^i(\cdot)$ where $\mathcal{L}(\cdot)$ returns the one-hot vector representation of each unique label. To prove Corollary 1.2, we show that $z_T^i(u) = z_T^i(v) \ \forall i$.

*Base case $t = 0$*: Since $\pi(u) = v$ under the isomorphic mapping, their labels must be equal: $\mathcal{L}^i(u) = \mathcal{L}^i(v) \ \forall i \Leftrightarrow z_0^i(u) = z_0^i(v) \ \forall i$.

*Induction hypothesis $t = k$*: Assume that $z_k^i(u) = z_k^i(v) \ \forall i$.

*Induction step*: We want to show that $z_{k+1}^i(u) = z_{k+1}^i(v) \ \forall i$. Let's revisit Equation 3:

$$z_{k+1}^i(u) = z_k^i(u) + \frac{1}{\mathbf{d}} \sum_{w \in \mathcal{N}(u)} z_k^i(w)$$

Because $z_k^i(u) = z_k^i(v)$, we have:

$$\frac{1}{\mathbf{d}} \sum_{w \in \mathcal{N}(u)} z_k^i(w) - \frac{1}{\mathbf{d}} \sum_{w \in \mathcal{N}(v)} z_k^i(w)$$

$$= \frac{1}{\mathbf{d}^{k+1}} \sum_{w_1 \in \mathcal{N}_{k-1}(u)} \deg(w_1)\mathcal{L}^i(w_1) + \frac{1}{\mathbf{d}^{k+1}} \sum_{w_2 \in \mathcal{N}_k(u)} \mathcal{L}(w_2)$$

$$- \frac{1}{\mathbf{d}^{k+1}} \sum_{\tilde{w}_1 \in \mathcal{N}_{k-1}(v)} \deg(\tilde{w}_1)\mathcal{L}^i(\tilde{w}_1) - \frac{1}{\mathbf{d}^{k+1}} \sum_{\tilde{w}_2 \in \mathcal{N}_k(v)} \mathcal{L}(\tilde{w}_2)$$

$$= \delta_z$$

This is to say that the (new) difference (if any) at the $(k+1)^{\text{th}}$ hop can only come from the labels of the new nodes reachable at $k$ hops away and the degrees of the nodes at $(k-1)$ hops away. This is because $z_t^i(w_2)$ at $k$ hops away include nodes that are $(k-1)$ hops away. The number of repetitions depends on the degree of each node at $(k-1)$ hops away. Since $\mathcal{N}_{k-1}(u) = \mathcal{N}_{k-1}(v)$ and $\mathcal{N}_k(u) = \mathcal{N}_k(v)$ along with the corresponding degrees, the difference $\delta_z = 0$, i.e. $z_{k+1}^i(u) = z_{k+1}^i(v)$. $\qquad\square$

Combining Corollary 1.1 and 1.2 proves the first part of Lemma 1.

**Proof of Lemma 1.2**   To ensure that: if $z_t^i(u) < z_t^i(v)$ then $z_{t+1}^i(u) < z_{t+1}^i(v)$ $\forall t < T$, we must discount the additive term of the newly reachable neighbors so that their sum cannot exceed the difference established in the previous iterations. We have:

$$\Delta = z_t^i(v) - z_t^i(u) \geq \frac{1}{\mathbf{d}^t}$$

$$\Delta' = \frac{1}{\mathbf{d}^{t+1}} \left( \sum_{w \in \mathcal{N}(u)} z_t^i(w) - \sum_{\tilde{w} \in \mathcal{N}(v)} z_t^i(\tilde{w}) \right) \leq \frac{1}{\mathbf{d}^{t+1}} \sum_{w \in \mathcal{N}_t(u)} \mathcal{L}^i(w) \leq \frac{\left|\mathcal{N}_t(u)\right|}{\mathbf{d}^{t+1}}$$

The first inequality holds because the minimum difference between $u$ and $v$ must come from last iteration. Otherwise, a deficit from earlier iterations has continued to grow larger.

With $M = \max_{u \in \mathcal{V}} \deg(u)$, we have: $\frac{\left|\mathcal{N}_t(u)\right|}{\mathbf{d}^{t+1}} \leq \frac{M^{t+1}}{\mathbf{d}^{t+1}}$. Therefore, we must make sure that:

$$\Delta > \Delta' \Leftrightarrow \frac{1}{\mathbf{d}^t} > \frac{M^{t+1}}{\mathbf{d}^{t+1}} \Leftrightarrow \mathbf{d} > M^{t+1}$$

For this to be true for any $t < T$: $\mathbf{d} > M^T$. Since this condition ensures that for all index $i$: if $z_t^i(u) < z_t^i(v)$ then $z_{t+1}^i(u) < z_{t+1}^i(v)$, it follows that: if $z_t(u) \neq z_t(v)$ then $z_t(u) \neq z_t(v)$. This proves the second part of Lemma 1. The implication of this result is that if the multisets of $u$'s and $v$'s neighbors are not equal at any hop, their final embeddings $z_T(u)$ and $z_T(v)$ are guaranteed to be different.

Next, we discuss the choice of $\mathbf{d}$ when $\mathbf{d} \leq M^T$. We have:

$$z_T^i(u) = \mathcal{L}^i(u) + \frac{c_1}{\mathbf{d}} + \frac{c_2}{\mathbf{d}^2} + ... + \frac{c_T}{\mathbf{d}^T}$$

where $(c_1, c_2, ..., c_T)$ are the terms discounted by $(\mathbf{d}, \mathbf{d}^2, ..., \mathbf{d}^T)$ respectively. With $\mathbf{d} = 1$, for a unique combination of $(c_1, c_2, ..., c_T)$, any permutation of this combination will still yield the same result of $z_T^i(u)$. This is not true for $\mathbf{d} > 1$. Therefore, the probability of collision with $\mathbf{d} > 1$ is much lower. However, it is hard to say for example whether $\mathbf{d} = 2$ or $\mathbf{d} = 3$ gives a lower probability of collision, as this depends on the underlying connectivity and label distribution of the graph. On the other hand, a very large $\mathbf{d}$ may cause numerical errors. In our setup, we choose $\mathbf{d} = 2$ for simplicity.

### A.1.2   COMPLEXITY AND COMPARISON OF TREE TYPES

We have described the hybrid tree traversal algorithm in Section 3.1.2. Here, we provide its pseudocode in Algorithm 1.

---

**Algorithm 1** Hybrid Tree Traversal

---

**Require:** Graph $\mathcal{G} = (\mathcal{V}, \mathcal{E})$, Root Node $r \in \mathcal{V}$
**Ensure:** Hybrid Tree $\mathcal{T}$ rooted at $r$, visiting each edge in $\mathcal{E}$ exactly once
 1: Enqueue($Q, r$)
 2: **while** $Q$ is not empty **do**
 3:     $u \leftarrow$ Dequeue($Q$)
 4:     **for** $v$ in of $\mathcal{N}(u)$ **do**
 5:         **if** $(u, v)$ not visited **then**
 6:             Add edge $(u, v)$ to $\mathcal{T}$
 7:             Add $v$ to $\mathcal{T}$
 8:             $(u, v)$.visited $\leftarrow$ True
 9:             Enqueue($Q, v$)
10: **return** $\mathcal{T}$

---

**Definition 8** (Chordless cycle). A chordless cycle $\mathcal{C}(\mathcal{V}_\mathcal{C}, \mathcal{E}_\mathcal{C})$ of a graph $\mathcal{G}(\mathcal{V}, \mathcal{E})$ is a simple cycle in which no edge connecting any two vertices of the cycle is not part of the cycle itself. In other words, $\forall (u, v) \in \mathcal{E}, u \in \mathcal{V}_\mathcal{C}, v \in \mathcal{V}_\mathcal{C}$, it must be that: $(u, v) \in \mathcal{E}_\mathcal{C}$.

With this definition, we prove the following lemma.

**Lemma 2**  In a hybrid tree, the number of duplicating nodes is equal to the number of chordless cycles in the original graph.

*Proof.* We first make two observations:

1. Because a hybrid tree visits all edges of the original graph at least once, for every chordless cycles, all of its edges have been visited. To successfully visit all the edges in a chordless cycle, at least one node in the cycle must be repeated. Therefore, the number of duplicating nodes $\geq$ the number of chordless cycles.
2. From Algorithm 1, we only visit an edge if it has not been visited. Hence, the current node can't visit its parent. Therefore, if this edge results in one duplicating node, it must form a chordless cycle which contains the duplicating node. This has to be a *new* chordless cycle. If this chordless cycle has been previously created, that would indicate that the current edge has already been visited, which is contradictory to the design of Algorithm 1. Therefore, for every duplicating node, there is one chordless cycle, i.e. the number of duplicating nodes $\leq$ the number of chordless cycles.

With these two observations, we can conclude that the number of duplicating nodes must be equal to the number of chordless cycles. $\square$

---

**Algorithm 2** Computation Tree Expansion

---

**Require:** $\mathcal{G} = (\mathcal{V}, E)$, Initial Root $r_0 \in \mathcal{V}$
**Ensure:** $\mathcal{T}$: A computation tree rooted at $r_0$
 1: **Global Constant:** MAX_HOPS: Maximum depth of tree

 2: **function** COMPUTATIONTREE($\mathcal{G}, r, n\_hops$)
 3:     Initialize $\mathcal{T}$ as an empty tree
 4:     **if** $n\_hops >$ MAX_HOPS **then**
 5:         **return** $\mathcal{T}$
 6:     **for** $u$ in $\mathcal{N}(r)$ **do**
 7:         Append COMPUTATIONTREE($\mathcal{G}, u, n\_hops + 1$) to $\mathcal{T}$
 8:     **return** $\mathcal{T}$

 9: **return** COMPUTATIONTREE($\mathcal{G}, r_0, 0$)

---

**The growth of Computation Tree.**  Due to the recursive addition of neighbors with no constraint on repetitions (see Algorithm 2), the number of nodes in a computation tree can grow exponentially.

Table 5: Summary of the statistics of the datasets

| Dataset | Size | Avg training $|\mathcal{V}|$ | Avg training $|\mathcal{E}|$ | Labels | Domain |
|---|---|---|---|---|---|
| ogb-molhiv | 100k | 18.97 | 23.55 | 119 | Molecules |
| ogb-code2 | 100k | 23.56 | 22.55 | 97 | Software |
| AIDS | 285k | 8.89 | 8.8 | 29 | Molecules |
| LINUX | 971k | 8.02 | 7.43 | Unlabeled | Software |
| IMDB | 288k | 14.35 | 74.67 | Unlabeled | Movies |

In order to visit all the nodes at least once, MAX_HOPS $\geq R$ where $R$ is the radius of the graph (the minium eccentricity). On average, the number of nodes in the computation tree will be $O(\bar{d}^R)$ where $\bar{d}$ is the average degree. In the best-case scenario, the nodes are visited in the ascending order of their degrees. This is because a node of higher degree appearing earlier near the top of the tree will be repeated more times, spawning more duplicates of its neighbors. This optimistic scenario provides the lower bound of the number of nodes in a computation tree: $\max_k\{(d_{k-1})^{R-k}\}$ where $d_{k-1}$ is the $(k-1)^{\text{th}}$ smallest degree. This lower bound can still be large.

## A.2 ADDITIONAL DETAILS ON EXPERIMENTAL SETUP

**Reproducibility.** Our code is available at: https://anonymous.4open.science/r/ged_distill-B177/ .gitignore.

### A.2.1 DATASETS

We use the following datasets (Table 5 provides a summary). We obtain AIDS, Linux and IMDB from Morris et al. (2020); Ranjan et al. (2022), the other datasets are from Hu et al. (2021) and Verma et al. (2025).

- *ogb-code2* (code2): This dataset contains a large collection of Abstract Syntax Trees (ASTs) derived from approximately 450,000 Python method definitions. Each graph corresponds to an AST, with nodes labeled from a fixed set of 97 categories that encapsulate various syntactic elements of the methods. These graphs are treated as undirected.
- *ogb-molhiv* (molhiv): The dataset consists of chemical compounds, where each graph represents a distinct molecule. The graph nodes correspond to atoms and are labeled according to their atomic numbers, while the edges represent the chemical bonds between atoms.
- *AIDS*: This dataset comprises graphs derived from the AIDS antiviral screening database, where each graph models the molecular structure of a chemical compound. The graphs are labeled to reflect meaningful chemical properties. They are relatively small, with each containing no more than 10 nodes.
- *LINUX*: This consists of program dependence graphs in which nodes represent individual statements and edges capture dependencies between them. Similar to the AIDS dataset, graph sizes are capped at 10 nodes. The dataset is also unlabeled Wang et al. (2012).
- *IMDB*: This unlabbeled dataset features ego-networks of actors and actresses who have co-starred in movies Yanardag & Vishwanathan (2015). Each graph represents the network of a single individual, with nodes corresponding to other actors/actresses and edges indicating shared film appearances.

### A.2.2 DETAILS ON HYPERPARAMETERS

A major advantage of TREEGEAR is that it has only a few hyperparameters. In the construction of the hybrid trees, we set the maximum hops reachable to be 3 for AIDS and LINUX, 5 for IMDB, and 10 for code2 and molhiv. This is influenced by the typical size of a graph in each dataset (see Table 5), rather than by an expensive hyperparameter search. This highlights the robustness of our method. On the other hand, the boosting factor $\delta$ for NBM is only involved during inference. Since any GED returned by TREEGEAR is an upperbound, we can try different values of and use the one that yield the minimum upperbound. As a rule of thumb, $\delta$ is set to be 0.7 for AIDS and 1.6 for others. For the hyperparameters of the backbone GNN, we use the default values recommended by Ranjan et al. (2022).

## A.3 Additional Experiments

Here we present additional experiments and results. First, we compare the ground-truth computation time of TREEGEAR and that of the baseline. Second, we report other metrics of interest not shown in the main paper. Third, we show that TREEGEAR can be integrated with other neural frameworks for GED prediction. Lastly, we perform a study to demonstrate the effect of the embedding budget on the performance of NBM.

### A.3.1 Additional Metrics

We evaluate TREEGEAR in further aspects. First, we show in Table 6 that the TED values requested by TREEGEAR can be computed efficiently, in contrast to the exponential time spent on calculating the GED values for other baselines. In terms of other performance metrics, Table 7 demonstrates that TREEGEAR achieves higher rank correlation (Spearman and Kendall) and competitive or superior top-$k$ precision ($k = 10$ and $k = 20$) compared to the GREED baseline.

Table 6: Ground-truth Computation Time (ms/pair). TREEGEAR is much faster on all datasets. This is because TREEGEAR only requires TED, which is polynomially computable compared with the NP-hard GED needed by the baselines. The GED computation time is taken from Ranjan et al. (2022) and Bommakanti et al. (2024).

|  | AIDS | LINUX | IMDB | code2 | molhiv |
|---|---|---|---|---|---|
| TREEGEAR (TED) | 3.51 | 2.91 | 389.81 | 34.00 | 17.77 |
| Baselines (GED) | 350.08 | 123.26 | 4479.47 | 781.51 | 13655.75 |

Table 7: Additional metrics. We report the Spearman's Rank Correlation Coefficient, Kendall's Rank Correlation Coefficient, and precision at top 10 and top 20 (p@10, p@20). TREEGEAR shows better results than GREED in almost all settings.

|  | Spearman | Kendall | p@10 | p@20 |
|---|---|---|---|---|
| **AIDS** | | | | |
| TREEGEAR | $0.975 \pm 0.011$ | $0.959 \pm 0.003$ | $1 \pm 0$ | $1 \pm 0$ |
| GREED | $0.938 \pm 0.001$ | $0.866 \pm 0.002$ | $0.8 \pm 0$ | $0.75 \pm 0$ |
| **LINUX** | | | | |
| TREEGEAR | $0.999 \pm 0.001$ | $0.998 \pm 0.001$ | $1 \pm 0$ | $1 \pm 0$ |
| GREED | $0.972 \pm 0.003$ | $0.931 \pm 0.007$ | $1 \pm 0$ | $1 \pm 0$ |
| **IMDB** | | | | |
| TREEGEAR | $1 \pm 0.000$ | $0.998 \pm 0.001$ | $1 \pm 0$ | $1 \pm 0$ |
| GREED | $0.99 \pm 0.001$ | $0.935 \pm 0.000$ | $1 \pm 0$ | $1 \pm 0$ |
| **code2** | | | | |
| TREEGEAR | $0.975 \pm 0.001$ | $0.889 \pm 0.002$ | $0.6 \pm 0.1$ | $0.817 \pm 0.058$ |
| GREED | $0.936 \pm 0.001$ | $0.801 \pm 0.001$ | $0.7 \pm 0.0$ | $0.7 \pm 0.050$ |
| **molhiv** | | | | |
| TREEGEAR | $0.988 \pm 0.001$ | $0.932 \pm 0.003$ | $0.9 \pm 0.000$ | $0.867 \pm 0.029$ |
| GREED | $0.988 \pm 0.001$ | $0.933 \pm 0.002$ | $0.733 \pm 0.058$ | $0.75 \pm 0.050$ |

### A.3.2 Generalizability to Other Architectures

We integrate TREEGEAR with other established neural frameworks for GED prediction, namely GEDGNN Piao et al. (2023), ERIC Zhuo & Tan (2022), and H2MN Zhang et al. (2021). Here integrating means that we only use these models as base models to train with TED. The TREEGEAR model and the original model use the same hyperparameteters for fairness. As observed in Table 8,

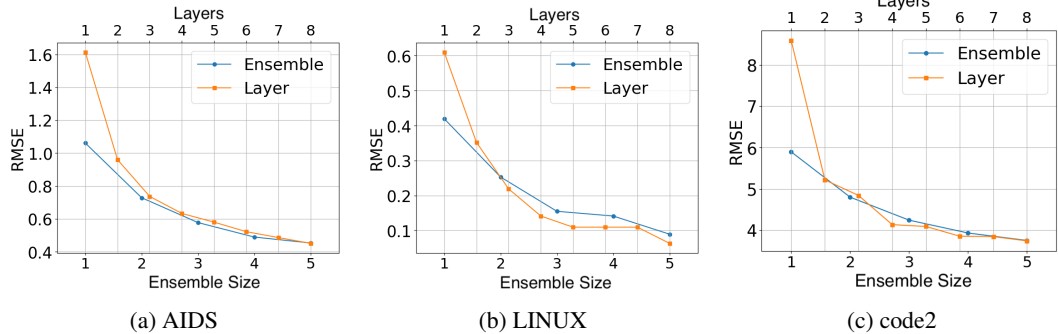

Figure 7: The effect of the node embeddings budget on the performance of NBM. We vary the numbers of node embeddings by fixing the number of submodels in the ensemble and changing the number of embedding layers used (orange line) as well as vice versa (blue line). In all three datasets, the orange line decreases at a faster rate than the blue line, suggesting the higher importance of the number of layers used for generating the embeddings.

TREEGEAR shows competitive or better performance than the base model even when TREEGEAR has no access to the ground-truth GED like the baselines. The result is mixed for H2MN. We believe that the reason is the very limited number of node-level embeddings provided by this model: H2MN only has two node-level convolution layers. This prevents the advantage of NBM in taking the minimum edit path. However, it should be emphasized again that the competitive performance of TREEGEAR against H2MN is achieved with no ground-truth labels. These results demonstrate the generalizability and applicability of TREEGEAR.

Table 8: The RMSE of TREEGEAR when integrated with other neural architectures on three datasets. Integrating means that TREEGEAR only use these models as base models to train with TED. TREEGEAR without being trained on GED values is better than the neural models which are trained on actual GED values.

| | AIDS | LINUX | code2 |
|---|---|---|---|
| TREEGEAR (H2MN as base) | $1.568 \pm 0.021$ | $0.513 \pm 0.027$ | $9.004 \pm 0.122$ |
| H2MN | $1.308 \pm 0.021$ | $1.041 \pm 0.024$ | $6.537 \pm 1.276$ |
| TREEGEAR (ERIC as base) | $1.431 \pm 0.221$ | $0.691 \pm 0.066$ | $4.183 \pm 0.102$ |
| ERIC | $2.721 \pm 0.000$ | $2.503 \pm 0.026$ | $33.633 \pm 0.000$ |
| TREEGEAR (GEDGNN as base) | $3.083 \pm 0.118$ | $2.419 \pm 0.061$ | $16.519 \pm 1.546$ |
| GEDGNN | $9.208 \pm 0.002$ | $5.185 \pm 0.000$ | $37.283 \pm 0.012$ |

### A.3.3 ADDITIONAL ABLATION STUDIES

We investigate the effect of the budget of embeddings on the performance of NBM. In TREEGEAR, we use two strategies to diversify the set of node embeddings available to NBM: (i) we use the embeddings output by each layer in the model, (ii) we train multiple submodels with different initializations on different data subsets. In total, this gives us $L \times E$ unique sets of embeddings where $L$ is the number of layers and $E$ is the size of the ensemble. In Figure 7, we look into the importance of each of these two factors on the performance of NBM. Specifically, we use all $L$ layers with different numbers of submodels and all $E$ submodels with an increasing number of layers. While both diversification strategies help bring down the RMSE, increasing the number of layers used for taking the embeddings gives more benefit, as the orange line has a steeper slope. This supports the notion that the limited number of layers in H2MN may have been an issue.

