# OpenReview forum: "TREEGEAR: Learning Graph Edit Distance with Zero Ground-truth Labels"
_ICLR.cc/2026/Conference — Submitted to ICLR 2026_

### Official Review · Reviewer_ZuCc · 2025-10-28

**Soundness:** 2
**Presentation:** 2
**Contribution:** 1
**Rating:** 2
**Confidence:** 2

**Summary:**

This paper provides a novel method to approximate the graph edit distance between two graphs. The proposed method uses tree edit distance as a proxy, making it scalable and more efficient compared to related methods.

**Strengths:**

The idea of training GNN on TED is interesting, allowing one to leverage the difficulties of supervised training of the GNN. The paper is relatively well written.

**Weaknesses:**

The underlying idea of training the GNN using the tree edit distance is interesting. This strategy shifts the problem to the selection of the trees from the graphs under study. This is a difficult task and requires some heuristics in order to properly select a pair of trees that cover the graph regions where structural differences are most likely to occur.
Such heuristics could be questionable, even though there are some proposed hypotheses.

The proposed strategy of approximating the GED through tree representations is not new. There are many related methods that are cited in the submitted paper, such as:
- Bause, F., Permann, C., & Kriege, N. M. (2024, August). Approximating the graph edit distance with compact neighborhood representations. In Joint European Conference on Machine Learning and Knowledge Discovery in Databases (pp. 300-318). Cham: Springer Nature Switzerland.
- Xu, M., & Chang, L. (2025). Graph Edit Distance Estimation: A New Heuristic and A Holistic Evaluation of Learning-based Methods. Proceedings of the ACM on Management of Data, 3(3), 1-24.

It is not clear why in the proposed formulation, the graph neural network does not directly predict the graph edit distance, but its output needs to be transformed with another function. It would have been more relevant to provide an end-to-end architecture.

The experimental analysis is in favor of the proposed method. However, the authors have chosen not to present all the comparable methods in all the tables. For instance, Table 2 providing the EMR compared the proposed method with only 3 neural methods, while Table 1 includes more competitive neural methods, such as H2MN and GraphEdx. Furthermore, it is not clear why the authors did not compare to non-neural methods, such as the 7 methods given in Table 1.
Moreover, it would have been relevant to consider the same datasets in all the tables.
At the end, it seems cherrypicking the best results for the proposed method.

**Questions:**

No further comments.

---

> ### Author Response · Authors · 2025-11-23
> **Rebuttal**
>
> Thank you for your comments!
>
> > W1: The proposed strategy of approximating the GED through tree representations is not new.
>
> We appreciate the reviewer's comment regarding the novelty of approximating GED through tree representations. The works cited by the reviewer (Bause et al. and Xu & Chang) are non-learning-based, algorithmic approaches. These approaches rely on fixed, hand-engineered rules and cannot leverage learned patterns to improve estimation quality for graph pairs belonging to the same distribution. Their performance is static and independent of the training data.
> On the other hand, our approach uses TED as a weak supervision signal to train a deep GNN. This allows our model to learn a structurally optimal embedding space that captures the nuances of the graph distribution, resulting in significantly improved estimation quality.
>
> We also clarify that the App-BMao algorithm by Xu & Chang is fundamentally different from ours. App-BMao is a search algorithm that operates on a search tree, not the graph-derived trees used in our method. More importantly, App-BMao only outperforms the neural baselines shown in our paper when the allowed computation time $t$ is sufficiently large, which may not be possible in practice.
>
> > W2: It is not clear why the GNN does not directly predict the graph edit distance, but its output needs to be transformed with another function. It would have been more relevant to provide an end-to-end architecture.
>
> **Response:** The choice of our two-stage architecture is a deliberate design decision driven by our training objective and the need for both accuracy and scalability:
> 1. Inconsistency of Direct Output due to Weak Supervision: Our GNN is distantly supervised using the TED. Consequently, the direct, raw output of the GNN is not guaranteed to be consistent with the true GED distribution. It may not capture the precise costs and path of the optimal structural transformation.
> 2. Leveraging Embeddings for Alignment: However, the TED supervision forces the GNN to learn a powerful, structurally-aware node embedding space. This learned embedding space is still highly informative and provides the crucial input needed for the next stage: the node-level alignment scheme (NBM). The NBM then uses these rich embeddings to infer a precise, physically realizable edit path, which in turn yields a final, accurate GED value. It not only achieves good in-distribution performance but also generalizes well across datasets (as shown in Figure 5), confirming that the model learns a transferable representation of structural affinity.
> 3. Comparison to End-to-End Baselines: We also emphasize that many of the neural baselines presented in our paper are end-to-end architectures. The inherent limitation of these approaches is their strict requirement for ground-truth GED labels, which are prohibitively expensive to compute. In contrast, our approach uses the scalable TED supervision. Crucially, as shown in Table 1, our TED-supervised, node-level approach defeats these end-to-end architectures without requiring any costly ground-truth GED labels.
>
> > W3: the authors have chosen not to present all the comparable methods in all the tables.
>
> The reason for not including every baseline in Table 2 (Exact Match Ratio - EMR) is twofold:
> 1. High Correlation with RMSE: The EMR metric is highly correlated with the Root Mean Square Error (RMSE) shown in Table 1. Table 1 is the primary performance table and does include all comparable baselines.
> 2. Focused Comparison: For Table 2, we deliberately selected the most competitive baselines from Table 1, especially for the structurally harder datasets (IMDB, Code2, and Molhiv). This allows the reader to focus on the comparison against the strongest competing methods, which is often the most critical evaluation point for novelty.
>
> We clarify the inclusion of datasets across the paper:
> * Section 4.1 (Main Results): The core results tables in this section include all five datasets (AIDS, LINUX, IMDB, Code2, and MolHIV) to ensure a complete evaluation against prior work and across different graph types.
> * Section 4.2 (Ablation Study): For the ablation study, we use only three out of five datasets to keep the analysis concise and manageable. This selection, however, was inclusive and strategic to represent the full spectrum of challenges:
>     * AIDS: A labeled, standard (old) benchmark.
>     * LINUX or IMDB: An unlabeled, standard (old) benchmark.
>     * Code2: A labeled, new dataset.
>
> We hope this clarifies our choices in presenting the experimental results.

---

### Official Review · Reviewer_oJeo · 2025-10-31

**Soundness:** 3
**Presentation:** 2
**Contribution:** 3
**Rating:** 4
**Confidence:** 4

**Summary:**

This paper proposes a framework to learn graph edit distance (GED) with graph neural networks (GNN). There are two novel contributions: (1) using polynomial-time tree edit distance (TED) for weak supervision instead of NP-hard GED, and (2) a _neighbor-biased mapper (NBM)_ to compute edit paths using TED-trained GNN to score pairs of nodes. Experimental results show superior performance than baselines trained with direct GED supervision. Further analysis includes out-of-distribution generalization, size generalization, robustness and ablation studies.

**Strengths:**

The paper presents two novel and clever ideas:
(1) weaker supervision from TED is enough to achieve accurate GED predictions,
providing an alternative to exact GED computations for training data generation, and,
(2) the NBM algorithm to produce node matchings using TED-trained GNN embeddings,
and computing GED from that (this second part is well known).

TreeGear achieves much better results than the baselines.

Ablations establish the importance of the NBM component and the HybridTree construction used therein.

**Weaknesses:**

My main concern with this paper is that it does not come full circle
in experimentally demonstrating the central motivation for using TED supervision
instead of GED.
Lower cost of training data generation would enable training on larger graphs
than existing methods,
but no experiment setup considers the graph-size scalability of TreeGear.

From Table 3, major gains are from NBM.
TED is worse than GED, which seems to be reasonable to compute for all experimental settings
considered in this paper.
Hence GED + NBM seems the more useful contribution.
Even for larger graphs approx GED with timeout + NBM might work better than TED.

The paper is imprecise and unpolished. Examples:
* Important parts such as hybrid tree construction were hard to understand.
* I found some examples of bad citations which do not fit well with the sentence. e.g. L040, L212
* Lots of citations are missing parentheses.

There are other works which compute GED with a node matching step [1, 2, 3, 4] which could be
baselines or at least should be discussed as related work,
but are missing (including from discussion, e.g. in L068-071) .

[1] MATA*: Combining learnable node matching with A* algorithm for approximate graph edit distance computation. In CIKM, 2023.

[2] Computing approximate graph edit distance via optimal transport. Proceedings of the ACM on Management of Data, 3(1):1–26, 2025.

[3] NOAH: Neural-optimized A* search algorithm for graph edit distance computation. In ICDE, 2021.

[4] GRAIL: Graph edit distance and node alignment using llm-generated code. In ICML, 2025.

**Questions:**

1. Can you show some results for larger graphs where exact GED computation is prohibitive?

2. How does TED compare with exact GED computation but with a reasonable timeout?

3. How does weakly supervised TreeGear compare with unsupervised GED methods, e.g. [1]?
(not published at a conference, so wouldn't change my opinion of this paper)

4. On L268, what is a "chordless cycle"? Please define for readers.

5. In Def. 5 what if the root node is deleted?

6. In Fig. 4 the input graph is symmetric wrt nodes 2 and 3. How does the hybrid tree construction break the symmetry to give an output graph which is asymmetric wrt nodes 2 and 3?

7. In Fig. 2, "trained GNN" would be better than "pretrained GNN".


[1] EUGENE: Explainable Structure-aware Graph Edit Distance Estimation with Generalized Edit Costs

---

> ### Author Response · Authors · 2025-11-23
> **Rebuttal: part 1**
>
> Thank you for your detailed comments!
>
> > W1 and Q1: Lower cost of training data generation would enable training on larger graphs than existing methods, but no experiment setup considers the graph-size scalability of TreeGear.
>
>  We thank the reviewer for raising the important question regarding the graph-size scalability of our method, TREEGEAR. Indeed, a core motivation for using TED as a supervision signal is to bypass the prohibitive cost of exact GED computation, making our approach highly applicable to large graphs.
>
> Our experimental setup and results directly demonstrate TREEGEAR's effectiveness in scenarios involving large and out-of-distribution graphs:
>
> * **Cross-Dataset Generalization to Larger Graphs (Figure 5.a):** We demonstrated that TREEGEAR maintains strong performance when trained on one dataset and tested on another. Most notably, when TREEGEAR was trained on the LINUX dataset but tested on the IMDB dataset - whose graphs are larger and significantly denser - TREEGEAR outperformed the baseline by the greatest margin. This result provides compelling evidence that our method learns a robust, transferable representation of structural affinity that scales well to more complex graph structures.
> * **Upper Bound Tightness on Very Large Graphs (Figure 5.b):** To further stress-test TREEGEAR on datasets where exact GED cannot be computed due to graph size, we evaluated its performance on the OGB-PPA dataset.
> Graphs in the OGB-PPA dataset are substantial, having over 2,000 edges on average, making exact GED calculation infeasible. We used the model trained on the smaller Code2 dataset to predict values for OGB-PPA. Crucially, because our method is founded on node alignment, the predicted value is guaranteed to be a provable upper bound on the true GED (a guarantee most other neural methods lack). By comparing our result against established algorithmic baselines that provide upper bounds, we have assessed performance based on the tightness of the returned upper bound. As shown in Figure 5.b, TREEGEAR achieves a much tighter upper bound than the algorithmic baselines, confirming its effectiveness and scalability for providing high-quality estimations on very large graphs.
>
> > W2 and Q2: GED + NBM seems the more useful contribution. Even for larger graphs approx GED with timeout + NBM might work better than TED.
>
> We appreciate the reviewer's suggestion that using GED + NBM might be more useful. We directly address this by comparing the performance, cost, and feasibility of GED vs. our TED-supervised approach.
>
> **1. Performance vs. Computational Cost:** While we acknowledge that using the gold-standard GED + NBM is often empirically superior to TED + NBM, we observe two critical facts. First, as demonstrated in Table 3, the performance improvement gained by using GED supervision is not always significant, proving the high quality and worth of the TED signal. Second, this marginal improvement comes at an exceptionally high computational cost. To illustrate this cost disparity, we refer to the computation time comparison (ms/pair) in Table 6:
>
> | | AIDS | LINUX | IMDB | code2 | molhiv |
> | :--- | :---: | :---: | :---: | :---: | :---: |
> | TREEGear (TED) | **3.51** | **2.91** | **389.81** | **34.00** | **17.77** |
> | Baselines (GED) | 350.08 | 123.26 | 4479.47 | 781.51 | 13655.75 |
>
> **2. Addressing GED with Timeout:** To directly evaluate the reviewer's proposed "GED with timeout" strategy, we conducted an experiment based on an equal time budget. We assumed that the number of GED examples that could be computed within the same time budget as our TED approach is proportional to the time ratio between the two methods. We restricted the training data for the "GED with timeout" model to: $\frac{\text{TED time}}{\text{GED time}} \times n$. As shown below, the TED-supervised pipeline consistently outperforms the GED with timeout approach on all five datasets. The time constraint not only limits the generalizability of the resulting model but also prevents us from employing critical tactics like training an ensemble of models on different subsets to diversify the embedding scheme for NBM.
>
> **RMSE of supervision by TED and GED with timeout**
> | | AIDS | LINUX | IMDB | code2 | molhiv |
> | :--- | :---: | :---: | :---: | :---: | :---: |
> | TED | **0.43** | **0.15** | **1.24** | **3.67** | **2.5** |
> | GED timeout | 1.41 | 0.98 | 6.83 | 5.98 | 3.18 |
>
> > W3.1: Important parts such as hybrid tree construction were hard to understand.
>
> We apologize that the explanation of the hybrid tree construction was not clear. We provide the detailed algorithm in Section A.1.2 of the Appendix. Due to the space constraints of the initial submission, we could not include it in the main paper. We agree that this is a critical component and will move this algorithm to the main paper in the revised version to significantly improve clarity and flow.
>
> > W3.2: Unfit citations
>
> Thank you for pointing these out. We'll check and fix them.

---

> > ### Author Response · Authors · 2025-11-23
> > **Rebuttal: part 2**
> >
> > > W4: Other works which compute GED with a node matching step
> >
> > We thank the reviewer for directing our attention to other relevant works. We want to clarify our engagement with this specific body of literature:
> > * Inclusion of [2]: We confirm that we have already included the unsupervised version of [2]: GEDGW as one of our non-neural baselines.
> > * Exclusion of [3] (Noah): We observed that this method has been outperformed by the GEDGNN baseline (a competitive neural approach we did include). Therefore, to keep our set of baselines concise and focused on the state-of-the-art, we elected not to include it as a separate baseline.
> > * Discussion of [1] and [4]: We agree that the works referenced as [1] and [4] are highly relevant. We will include a dedicated discussion of these two papers in the related work section of our revised manuscript to ensure comprehensive context.
> >
> > > Q3: How does weakly supervised TreeGear compare with unsupervised GED methods?
> >
> > **Response:** While we didn't include EUGENE, which has not yet been published, we have already included another unsupervised method GEDGW, as noted above.
> >
> > > Q4: On L268, what is a "chordless cycle"? Please define for readers.
> >
> > **Response:** We give the formal definition in section A.1.2 in the Appendix.
> >
> > > Q5: In Def. 5 what if the root node is deleted?
> >
> > **Response:** This is indeed an interesting point. The original TED paper [5] doesn't explicitly specify the handling of the root deletion. However, given that the overall algorithm uses the concept of distance between ordered forests, we infer that an ordered forest is the logical result of deleting the root.
> > [5] Kaizhong Zhang and Dennis Shasha. Simple fast algorithms for the editing distance between trees and related problems. SIAM journal on computing, 18(6):1245–1262, 1989.
> >
> > > Q6: In Fig. 4 the input graph is symmetric wrt nodes 2 and 3. How does the hybrid tree construction break the symmetry to give an output graph which is asymmetric wrt nodes 2 and 3?
> >
> > We refer the reviewer to the detailed algorithm of hybrid tree construction in Section A.1.2 of the Appendix. The algorithm introduces dummy, duplicating nodes, which may break the symmetry of the original graph.
> >
> > > Q7: In Fig. 2, "trained GNN" would be better than "pretrained GNN".
> >
> > **Response:** We'll change "pretrained GNN" to "TED-trained GNN" for clarity.

---

### Official Review · Reviewer_RYH1 · 2025-11-01

**Soundness:** 3
**Presentation:** 2
**Contribution:** 1
**Rating:** 6
**Confidence:** 3

**Summary:**

This paper proposes a framework for learning the graph edit distance (GED) and alleviates computational complexity by learning the tree edit distance (TED) (i.e., a weakly supervised GED). Specifically, to leverage TED, the authors propose a method to transform graph structures into tree structures in the following order: 1) root selection, 2) hybrid tree construction, and 3) a Neighbor-Biased Mapper (NBM). GED is then computed by aligning the mapped nodes using four operators: node substitution, edge deletion, node insertion, and edge insertion. A GNN is trained to predict the edit cost after selecting the final matching path. The proposed approach is efficient in terms of computational complexity and exhibits strong generalization across diverse graph domains. Experiments on graph edit distance prediction tasks show that the proposed method outperforms prior approaches.

**Strengths:**

* The idea of weakly supervised GED leveraging TED is interesting and demonstrates its efficiency.
* The ablation study is well-designed to demonstrate the effectiveness of each component in the proposed method.
* The proposed method can generalize to diverse domains and offers interpretability.

**Weaknesses:**

* My primary concern is that the proposed method is evaluated only on graph edit distance prediction tasks, which limits its overall contribution and scope of application. I recommend conducting experiments on a wider range of tasks and comparing it against other representation learning methods.
* The performance heavily depends on the quality and diversity of the node embeddings fed into the NBM, preventing the NBM from fully leveraging its advantage by selecting the minimum edit path.

**Questions:**

* The performance tendency two molecular datasets are different (i.e., ADIS and molhiv). Could you explain why this difference occurs?
* Can learning the graph edit distance mitigate the oversmoothing problem?

---

> ### Author Response · Authors · 2025-11-23
> **Rebuttal: Part 1**
>
> Thank you for your comments!
>
> > W1 and Q2: The proposed method is evaluated only on GED prediction tasks, which limits its overall contribution and scope of application. Can learning the GED mitigate the oversmoothing problem?
>
> **Response:** Thank you for your comment. While the paper focuses on the challenging task of Graph Edit Distance (GED) estimation, we emphasize that this is not a limiting factor but rather the foundation for numerous downstream applications. As a generic and powerful measure of structural difference, GED is crucial across various domains:
> * Distance-Based Learning: GED is the essential distance metric that empowers fundamental algorithms like $k$-Nearest Neighbors ($k$-NN), Support Vector Machines (SVMs), and clustering for graph data [1].
> * Graph Retrieval and Mining: It is fundamental for retrieving structurally similar graphs from large databases and for pattern mining in graph sets [2].
> * Real-World Fields: These applications span biology and chemistry to computer vision and malware detection [3, 4]. Indeed, the challenging nature of GED estimation has led to numerous prior works dedicated solely to solving this NP-hard problem.
>
> Our method's output extends beyond a single distance value, offering two critical advantages for application:
> * **Interpretable Edit Paths:** Our approach produces an interpretable edit path, which is crucial for domains like molecular research where understanding how one structure transforms into another is as important as the final difference value.
> * **Network Alignment:** Since the edit path is derived from a node-level alignment (via the Neighbor-biased Mapper, NBM), our method is immediately applicable to the problem of network alignment and graph matching [5].
>
>
> **Regarding Oversmoothing:** Regarding the question of whether learning GED can mitigate the oversmoothing problem, we note that oversmoothing is typically described as a node-level phenomenon that impacts the distinctiveness of node embeddings in deep GNNs. In contrast, GED estimation is a graph-level problem that inherently involves the comparison of pairs of graphs. Similar to prior works in this area, the scope of our paper focuses on providing an accurate and efficient solution to the core NP-hard GED estimation problem.
>
> To further demonstrate the foundational accuracy and applicability of our method, we evaluate it on the Graph Isomorphism Test, a special case of GED. Per our method, graphs are isomorphic if the predicted GED is 0 and non-isomorphic otherwise. Our evaluation reveals that the method achieves **perfect accuracy (100%)** on the Graph Isomorphism Test across all test sets presented, validating its precision in structural difference assessment.
>
> [1] Gao, Xinbo, et al. "A survey of graph edit distance." Pattern Analysis and applications 13.1 (2010): 113-129.
>
> [2] Chang, Lijun, et al. "Accelerating graph similarity search via efficient ged computation." IEEE Transactions on Knowledge and Data Engineering 35.5 (2022): 4485-4498.
>
> [3] Garcia-Hernandez, et al. "Ligand-based virtual screening using graph edit distance as molecular similarity measure." Journal of chemical information and modeling 59.4 (2019): 1410-1421.
>
> [4] Kinable, Joris, and Orestis Kostakis. "Malware classification based on call graph clustering." Journal in computer virology 7.4 (2011): 233-245.
>
> [5] Saxena, Shruti, and Joydeep Chandra. "A survey on network alignment: approaches, applications and future directions." Proc. 33rd Int. Joint Conf. Artif. Intell.(IJCAI). 2024.

---

> > ### Author Response · Authors · 2025-11-23
> > **Rebuttal: Part 2**
> >
> > > W2: The performance heavily depends on the quality and diversity of the node embeddings fed into the NBM, preventing the NBM from fully leveraging its advantage by selecting the minimum edit path.
> >
> > **Response:** We agree with the reviewer's assessment that the performance of classical graph matching frameworks like NBM is heavily reliant on the quality and diversity of the input node embeddings. This dependency is precisely why the combination of NBM with the rich, learned representations provided by deep GNNs is highly beneficial, as clearly validated by our empirical results. To further ensure the NBM can effectively select the minimum edit path by leveraging diverse information, we employed two key practical strategies:
> > 1. We train an ensemble of $M$ models, each on a different subset of the training data. This ensures that the base representations are robust and capture varied aspects of the structural distribution.
> > 2. Rather than using only the final layer embedding, we leverage the embeddings from every layer ($L$ in total) of each trained GNN model. This approach provides the NBM with access to node information at different $k$-hop neighborhoods.
> >
> > In combination, this means the NBM is provided with a total of $M \times L$ diverse embedding schemes. This significantly enhances the diversity and information richness available to the NBM, enabling a more robust and informed decision for deriving the optimal node correspondence and subsequent minimum edit path. **We have provided a detailed ablation and analysis of the contribution of this embedding strategy in Section A.3.3 of our Appendix.**
> >
> > > Q3: The performance tendency two molecular datasets are different (i.e., ADIS and molhiv). Could you explain why this difference occurs?
> >
> > **Response:** Thank you for your observation. While both reside in the molecular domain, the difference in performance is attributable to key differences between the datasets, which interact differently with our GNN architecture:
> > * The graphs in Molhiv are significantly larger and typically denser than those in AIDS. This increased complexity demands a more expressive model to capture global structural differences effectively.
> > * The most critical difference lies in the complexity of the node features: AIDS contains only 29 unique node labels while Molhiv contains 119 unique node labels, the highest count among all five datasets we tested. Our GNN architecture uses a default hidden dimension of 64, a parameter size inherited from the GREED paper [6]. That paper's evaluation was only conducted on datasets with either a smaller number of unique labels (AIDS: 29) or unlabeled graphs (LINUX, IMDB: input dimension of 1). In the case of Molhiv, the input feature dimension (119) is significantly larger than the hidden dimension. It is highly probable that having the hidden dimension smaller than the input dimension resulted in some information loss during the initial feature projection and subsequent message passing. This information compression may have prevented our method from achieving the top performance on Molhiv, unlike on AIDS. Despite this limitation posed by the default architecture setting, we emphasize that our method, even with the constrained hidden dimension, still achieves competitive results against baselines that have full access to the ground-truth Graph Edit Distance (GED) labels on the challenging Molhiv dataset. This underscores the robustness of our TED-supervised training and NBM mapping approach.
> >
> > [6] Ranjan, Rishabh, et al. "Greed: A neural framework for learning graph distance functions." Advances in Neural Information Processing Systems 35 (2022): 22518-22530.

---

### Official Review · Reviewer_PZW2 · 2025-11-01

**Soundness:** 1
**Presentation:** 2
**Contribution:** 2
**Rating:** 2
**Confidence:** 4

**Summary:**

This paper proposes TREEGEAR, a framework for learning to approximate the Graph Edit Distance (GED) without relying on ground-truth GED supervision, which is costly to compute due to the NP-hard nature of GED. The key idea is to extract (so called hybrid) rooted trees from graphs and use tree edit distance (TED)—which is polynomial-time computable—as a proxy supervision signal. A GNN is trained on pairs of such trees to produce node-level embeddings that capture structural similarity. At inference time, node embeddings are used with a neighbor-biased mapping (NBM) procedure to produce a node correspondence, from which an edit path and corresponding GED upper-bound can be derived.

**Strengths:**

The motivation is well-founded: current neural GED approximators rely on expensive ground-truth GED labels, restricting scalability and practical applicability. The idea of replacing GED supervision with a relatively easily available tree distance based proxy is conceptually appealing.

**Weaknesses:**

As far as I understand, the method uses tree edit distance as a proxy signal to train node embeddings, and then applies an injective node mapping strategy (NBM) to derive a node correspondence and subsequently an edit path. While the choice of tree representation (e.g., hybrid trees vs. BFS trees) may affect performance to some extent, the overall inference procedure, node matching based on pairwise node affinity (resulting in an interpretable edit path),  is already well-established. Thus, beyond the use of TED-supervised training, I do not see anything new here.

Now, my primary concern is the role and necessity of the TED-supervised training process itself. If the final node alignment depends only on the relative ordering of node similarity scores (as is the case with NBM), then it is unclear what benefit the neural training stage provides. In fact, untrained GNNs with distance based downweighting can already capture neighborhood-aware structural information. Then what is the model actually learning when trained to regress tree edit distance?

Lastly,  AIDS and LINUX benchmarks used in the experiments are known to have structural leakage [1], meaning that graphs in the test split are often isomorphic to the training graphs. As such any evaluation currently done on these datasets is unreliable.


[1] Position: Graph Matching Systems Deserve Better Benchmarks. In Forty-second International Conference on Machine Learning Position Paper Track.

**Questions:**

Quesitons are mostly based off the mentioned weaknesses.
- The NBM-based inference relies primarily on the relative ordering of node similarity scores. Given this, what is the justification for training the GNN to regress absolute TED values, rather than using an untrained or self-supervised embedding scheme?

- Beyond substituting GED labels with TED supervision, are there additional conceptual or methodological contributions that I may be overlooking?

- Several of the evaluation datasets (e.g., AIDS, LINUX) are known to exhibit structural leakage, which can inflate performance metrics. How do the authors interpret their results in light of this?

---

> ### Author Response · Authors · 2025-11-23
> **Rebuttal: Part 1**
>
> Thank you for your comments! Please see our point-wise rebuttal.
>
> > W2 and Q1: What is the justification for training the GNN to regress absolute TED values, rather than using an untrained or self-supervised embedding scheme?
>
> **Response:** We appreciate the reviewer's critical point on the necessity and role of the TED-supervised training process. While it is true that untrained GNNs can capture localized structural information, training the GNN to regress the absolute Tree Edit Distance (TED) value forces the model to learn a globally consistent, topologically aware embedding space optimized for the task of structural difference quantification. To directly address the reviewer's concern, we have tested the performance using the embeddings derived from a simple untrained sum convolution (chosen to inherently preserve node degree information). The results clearly demonstrate the benefit of our approach of TED supervision.
>
> As shown in the tables below, our TED-supervised method yields superior results compared to the untrained embedding scheme in terms of both Root Mean Square Error (RMSE) and Exact Match Ratio (EMR). This performance gap confirms that the model is learning more than simple local, neighborhood-aware structural properties; it is learning a representation that is necessary for accurate graph-level distance estimation.
>
> **RMSE of TreeGEAR and Untrained Sum Convolution**
> | | AIDS | LINUX | IMDB | code2 | molhiv |
> | :--- | :---: | :---: | :---: | :---: | :---: |
> | **TreeGEAR (Ours)** | **0.43** | **0.15** | **1.24** | **3.67** | **2.5** |
> | Sum-Conv | 1.36 | 0.79 | 4.43 | 8.11 | 4.51 |
>
> **EMR of TreeGEAR and Untrained Sum Convolution**
> | | AIDS | LINUX | IMDB | code2 | molhiv |
> | :--- | :---: | :---: | :---: | :---: | :---: |
> | **TreeGEAR (Ours)** | **88.8 %** | **99.5 %** | **97.1 %** | **16.2 %** | **24.7 %** |
> | Sum-Conv | 86.2 % | 50.4 % | 86.3 % | 2.3 % | 7.3 % |
>
> ---
>
> > W1 and Q2: Beyond substituting GED labels with TED supervision, are there additional conceptual or methodological contributions that I may be overlooking?
>
> **Response:** We appreciate the reviewer's query regarding the additional novelty of our approach beyond the TED supervision.
> * **Scalable and Practical Setting:** First, we would like to emphasize that the success of the TED-supervised training, as demonstrated by the empirical results above, is in itself a significant contribution. Moreover, the use of TED is a critical conceptual advance because, unlike the NP-hard GED, TED is polynomial-time computable. This bypasses the biggest practical hurdle in neural GED approaches: the prohibitively costly acquisition of ground-truth GED labels. This makes our approach scalable and practically feasible for large-scale training.
> * **Theory-grounded Tree Generation:** Second, our contribution includes the careful design of the hybrid tree. This design is specifically engineered to preserve the structural semantics crucial for the message-passing mechanism of a GNN. Our Hybrid Tree construction explicitly preserves the neighborhood for all original, non-duplicated graph nodes. We also provide a theoretical guarantee on the number of duplicated nodes, ensuring efficient and structurally sound representation (detailed in Section 3.1.2).
> * **Generalization:** Third, we propose a shift in how the neural GED problem is addressed. Instead of the prevalent graph-level regression problem view, we frame it as a node-level structural mapping problem using the Neighbor-biased Mapper (NBM). This approach not only helps us to achieve better results on in-distribution test sets but, more importantly, it manages to generalize well to out-of-distribution datasets, especially those with large graphs. We highlight this in Figure 5, where we train on one dataset and test on another.

---

> > ### Author Response · Authors · 2025-11-23
> > **Rebuttal: Part 2**
> >
> > > W3 and Q3: Several of the evaluation datasets (e.g., AIDS, LINUX) are known to exhibit structural leakage, which can inflate performance metrics. How do the authors interpret their results in light of this?
> >
> > **Response:** We thank the reviewer for raising the concern regarding potential structural leakage in standard GED datasets like AIDS and LINUX.
> > * First, we want to point out that AIDS and LINUX have been the de facto standard benchmarks and are widely used in every preceding neural GED paper. To ensure a fair and comprehensive comparison against the state-of-the-art baselines (as demonstrated in our results tables), we felt the necessity of including these two datasets in our experimental setup.
> > * Second, we'd like to note that a key motivation for employing neural approaches to solve algorithmic problems is their capacity to leverage in-distribution patterns to provide superior, learned estimations. Therefore, while excessive leakage is problematic, some degree of structural similarity between the training and test sets is inherently acceptable within the scope of distribution-sensitive machine learning.
> >
> > However, we agree with the reviewer that the current GED benchmark landscape has limitations. To proactively address the potential for inflated metrics due to leakage and to provide a more objective evaluation, we have introduced two additional, distinct datasets into our experiments: Code2 and Molhiv. Most importantly, to directly mitigate the impact of structural leakage, we performed a stringent cross-dataset generalization test. In this evaluation setting, we trained our model on one dataset but tested it on another (e.g., training on LINUX but testing on IMDB). As detailed in **Figure 5 of our paper (and mentioned in the response to Q2)**, our method maintains strong performance even under this out-of-distribution, cross-dataset setup. This generalization capability provides strong evidence that our model is learning fundamental and transferable structural affinities, rather than merely memorizing or exploiting dataset-specific structural patterns.

---

### Author Response · Authors · 2025-11-27

Dear AC, and Reviewers,

We are grateful for the reviewers' detailed and valuable feedback. Please let us know whether our responses appropriately address your concerns and whether there are any remaining questions that we can address before the end of the discussion period (ending in a week).  Our response is summarized as follows:

1. Necessity of TED Supervision
* Training the GNN on TED is crucial because it forces the model to learn a globally consistent, topologically aware embedding space.
* Empirical Proof: Direct comparison against an untrained Sum-Convolution GNN showed a significant performance gap across all five datasets.
* The cost disparity between TED and GED supervision is exceptionally high (Table 6).
* An experiment comparing TED supervision with a restricted "GED with timeout" showed that the TED-supervised pipeline consistently outperforms the timeout approach on all five datasets.
2. Conceptual and Methodological Novelty
* Scalability & Practicality: The use of TED is a critical conceptual advance because, unlike the NP-hard GED, TED is polynomial-time computable. This bypasses the biggest practical hurdle for neural methods.
* Unlike static, hand-engineered algorithms which used tree approximations before, TreeGEAR learns an embedding space that adapts to the graph distribution, resulting in significantly improved estimation quality.
* Theory-Grounded Structure: The approach includes the careful design of the Hybrid Tree construction to preserve structural semantics and includes a theoretical guarantee on the number of duplicated nodes (Sec 3.1.2).
* Node-Level Framing: The method shifts the problem from graph-level regression to a node-level structural mapping problem using the Neighbor-biased Mapper (NBM). This helps the model generalize well to out-of-distribution and large graphs (Fig 5).
* Embedding Diversity: To ensure NBM selects the minimum edit path robustly, the input includes: every layer ($L$) from an ensemble of $M$ models, providing $M \times L$ diverse representations.
3. Robustness, Leakage, and Generalization
* While standard benchmarks (AIDS, LINUX) were retained for state-of-the-art comparison, new datasets (Code2, Molhiv) were introduced for a more objective evaluation.
* Cross-Dataset Generalization: The model demonstrated strong performance in out-of-distribution tests (Fig 5).
* Graph Size Scalability: Cross-dataset tests showed TreeGEAR performing strongest when generalizing to larger, denser graphs (LINUX $\rightarrow$ IMDB). Evaluation on the massive OGB-PPA graphs confirms that TreeGEAR provides a tighter provable upper bound on the true GED (Fig 5).
* Isomorphism Test: The method achieved perfect accuracy (100%) on the Graph Isomorphism Test across all test sets, validating its precision.
* By producing an interpretable edit path from an injective node mapping, our method shows promising applications in Network Alignment and Molecular Research.

Regards,

Authors

---

### Meta-Review · Area_Chair_56LG · 2026-01-17

**Summary:**

1. The rationale for training the GNN to regress absolute TED values, given that NBM-based inference depends mainly on the relative order of node similarity scores—instead of using an untrained or self-supervised embedding approach.
2. No experiments are conducted to investigate the scalability of TreeGear on large-scale graphs.
3. Selecting tree pairs that target graph regions with the highest likelihood of structural differences is a challenging task requiring heuristics, whose validity remains questionable despite existing hypothesized justifications.
4. The authors’ inconsistent inclusion of comparable neural/non-neural methods across tables and failure to use uniform datasets raise concerns of cherry-picking favorable results.
5. There are some citation formatting errors and unclear definitions.

**Reviewer Concerns:**

Addressed concerns:
1. The performance tendency two molecular datasets are different (i.e., ADIS and molhiv). (From reviewer RYH1)
2. The definition of "chordless cycle". (From reviewer oJeo)

Still outstanding concerns:
1.The rationale for training the GNN to regress absolute TED values, given that NBM-based inference depends mainly on the relative order of node similarity scores—instead of using an untrained or self-supervised embedding approach. (From reviewer PZW2)
2. No experiments are conducted to investigate the scalability of TreeGear on large-scale graphs. (From reviewer oJeo)
3. Selecting tree pairs that target graph regions with the highest likelihood of structural differences is a challenging task requiring heuristics, whose validity remains questionable despite existing hypothesized justifications. (From reviewer ZuCc)
4. The authors’ inconsistent inclusion of comparable neural/non-neural methods across tables and failure to use uniform datasets raise concerns of cherry-picking favorable results. (From reviewer ZuCc)

**Reviewer Scores:**

Reviewers RYH1 and oJeo may retain their current ratings or add one point, as some of their concerns have been addressed in the authors' rebuttal.
Reviewers PZW2 and ZuCc may still hold their current ratings cause their main concerns are still not solved in the authors' rebuttal.

---

### Decision · Program_Chairs · 2026-01-26

Reject